# Integrated wafer-scale ultra-flat graphene by gradient surface energy modulation

Xin Gao[1,2,3,8], Liming Zheng[1,2,8], Fang Luo[4,8], Jun Qian[1,2,8], Jingyue Wang[1], Mingzhi Yan[2,5], Wendong Wang[6], Qinci Wu[1,2], Junchuan Tang[1], Yisen Cao[2], Congwei Tan[1], Jilin Tang[1,2,3], Mengjian Zhu[4]✉, Yani Wang[1,2], Yanglizhi Li[1,2,3], Luzhao Sun[2], Guanghui Gao[2,5], Jianbo Yin[2], Li Lin[2,7], Zhongfan Liu[1,2,3], Shiqiao Qin[4]✉ & Hailin Peng[1,2,3]✉

The integration of large-scale two-dimensional (2D) materials onto semiconductor wafers is highly desirable for advanced electronic devices, but challenges such as transfer-related crack, contamination, wrinkle and doping remain. Here, we developed a generic method by gradient surface energy modulation, leading to a reliable adhesion and release of graphene onto target wafers. The as-obtained wafer-scale graphene exhibited a damage-free, clean, and ultra-flat surface with negligible doping, resulting in uniform sheet resistance with only ~6% deviation. The as-transferred graphene on $SiO_2$/Si exhibited high carrier mobility reaching up ~10,000 cm$^2$ V$^{-1}$ s$^{-1}$, with quantum Hall effect (QHE) observed at room temperature. Fractional quantum Hall effect (FQHE) appeared at 1.7 K after encapsulation by h-BN, yielding ultra-high mobility of ~280,000 cm$^2$ V$^{-1}$ s$^{-1}$. Integrated wafer-scale graphene thermal emitters exhibited significant broadband emission in near-infrared (NIR) spectrum. Overall, the proposed methodology is promising for future integration of wafer-scale 2D materials in advanced electronics and optoelectronics.

The integration of two-dimensional (2D) materials into current silicon technology may embed high mobility, dangling band-free interface, atomic-scale channel size into practical electronic and optoelectronic devices[1–4]. Note that, an essential prerequisite is transferring 2D materials from their growth substrates to industrial wafers. Yet, a significant gap still exists in transfer method, which frustrates the recent progress in wafer-scale single-crystal growth of 2D materials[5–10]. Conventionally, wet transfer methods commonly use poly(methyl methacrylate) (PMMA) film as a transfer medium to support 2D materials when separated from the growth substrate and scooped from the liquid

surface to a target substrate[11,12]. For graphene, the wet transfer methods may significantly degrade its properties due to introduced cracks, wrinkles, polymer contaminations, and water doping by water adsorptions on the graphene surface[13–18]. These factors act as extra scattering centers to limit carrier mobility and decrease device performance. To overcome these issues, previous attempts have shown that the optimization of PMMA and replacement of PMMA with small molecules or other polymers would facilitate clean graphene transfer[19–27], conformal contact with the target substrate may reduce the formation of cracks and wrinkles[28–31], and the development of dry transfer methods may

---

[1]Center for Nanochemistry, Beijing Science and Engineering Center for Nanocarbons, Beijing National Laboratory for Molecular Sciences, College of Chemistry and Molecular Engineering, Peking University, Beijing 100871, P. R. China. [2]Beijing Graphene Institute, Beijing 100095, P. R. China. [3]Academy for Advanced Interdisciplinary Studies, Peking University, Beijing 100871, P. R. China. [4]College of Advanced Interdisciplinary Studies & Hunan Provincial Key Laboratory of Novel Nano-Optoelectronic Information Materials and Devices, National University of Defense Technology, Changsha, Hunan 410073, China. [5]School of Chemical Engineering & Advanced Institute of Materials Science, Changchun University of Technology, Changchun 130012, P. R. China. [6]School of Physics and Astronomy, University of Manchester, Manchester M13 9PL, UK. [7]School of Materials Science and Engineering, Peking University, Beijing 100871, P. R. China. [8]These authors contributed equally: Xin Gao, Liming Zheng, Fang Luo, Jun Qian. ✉e-mail: zhumengjian11@nudt.edu.cn; sqqin8@nudt.edu.cn; hlpeng@pku.edu.cn

diminish water doping by preventing the submersion of target substrate in liquids[32–39]. However, no method has so far entirely solved these issues, and most approaches are not compatible with high-volume semiconductor technology at the wafer level[40,41].

Here, we designed a multi-functional tri-layer transfer medium with gradient surface energy distribution, according to the thin-film adhesion theory that transfer of thin film from one layer to another layer is mainly dominated by the difference in surface energy of each layer[42,43]. In this case, the higher surface energy of a target substrate, the better it serves as the thin film 'acceptor' due to the better wetting and larger adhesion strength at interface[44]. Thus, the surface energies of transfer medium and target substrates should be engineered to ensure reliable adhesion and release[45], critical features for securing the wafer-scale 2D materials integration[1,2,46]. As a result, the gradient surface energy (GSE) modulation approach conduced to the integration of 4-inch single-crystal ultra-flat graphene onto silicon wafers. The transferred graphene wafer maintained its flatness, exhibiting intact and clean surfaces with negligible water doping. Consequently, the resulting wafer-scale graphene illustrated a uniform sheet resistance of only ~6% deviation over a 4-inch area. The as-transferred graphene on SiO$_2$/Si exhibited outstanding electrical performances with smaller Dirac points and much higher carrier mobility (~10,000 cm$^2$ V$^{-1}$ s$^{-1}$) at room temperature when compared to conventional wet transfer (~2000 cm$^2$ V$^{-1}$ s$^{-1}$). The quantum Hall effect (QHE) was also observed at room temperature in graphene transferred on SiO$_2$/Si, and fractional quantum Hall effect (FQHE) was recorded at 1.7 K in transferred graphene encapsulated by h-BN with high mobility reaching ~280,000 cm$^2$ V$^{-1}$ s$^{-1}$. Furthermore, the integrated thermal emitter arrays fabricated on a 4-inch graphene/silicon wafer showed significant emission with a broad spectrum in NIR region.

## Results

### Design of wafer-scale graphene integration

To minimize the adverse effects of grain boundaries and wrinkles on the charge carrier mobility, single-crystal ultra-flat graphene films were grown on 4-inch Cu(111)/sapphire wafers and the details can be found in Methods. A multi-functional tri-layer transfer medium was designed to support the wafer-scale graphene during transfer (Fig. 1a, b, Supplementary Figs. 1 and 2). The bottom layer of small molecules (borneol) was adsorbed on the graphene to reduce the surface energy of graphene, as well as working as a buffer layer to prevent direct contaminations caused by the upper PMMA layer (Fig. 1b). The PMMA layer ensured the integrity of graphene during transfer (Supplementary Fig. 3), and the topmost polydimethylsiloxane (PDMS) layer served as a self-supporting layer, allowing dry transfer of graphene and preventing water doping (Fig. 1a, b).

Most importantly, the surface energy gradually decreased from the destination SiO$_2$/Si wafer to the topmost PDMS layer (Fig. 1b, c), whose surface energy was calculated by measuring the contact angles based on Owen-Wendt and Young's equations[28,44] (Supplementary Table 1 and 2, Supplementary Fig. 4). The film with low surface energy tends to adsorb strongly on the substrate with high surface energy, according to the thin-film adhesion theory[31]. The driving force for wetting the interface is the spreading coefficient $\lambda_{AB}$:

$$\lambda_{AB} = \gamma_B - \gamma_A - \gamma_{AB} \tag{1}$$

where $\gamma_B$ and $\gamma_A$ are the surface energies of phase B (the adherend) and A (the adhesive), respectively; and $\gamma_{AB}$ is the interface energy between phase A and B. The fracture strength $\sigma_f$ of the interface is related to $\lambda_{AB}$ by:

$$\sigma_f = \frac{K_m}{1 - \frac{\lambda_{AB}}{\gamma_B}} = \frac{K_m \gamma_B}{\gamma_A + \gamma_{AB}} \approx \frac{K_m \gamma_B}{\gamma_A} \tag{2}$$

where $K_m$ is a function of the mechanical properties. Considering that $\gamma_{AB} \ll \gamma_A$, the fracture strength $\sigma_f$ is proportional to the surface energy ratio of the adherend to the adhesive ($\gamma_B/\gamma_A$).

For the wafer-scale graphene transfer, both the reliable adhesion and release of graphene film are critical, which determine the integrity of transferred wafer-scale graphene. Since the surface energy of SiO$_2$/Si ($\gamma_1$) was much larger than that of graphene/borneol ($\gamma_2$), the complete wetting and reliable adhesion of graphene to the SiO$_2$/Si wafer was facilitated (Fig. 1b, c). Moreover, the very small surface energy of PDMS ($\gamma_4$) that is close to PMMA ($\gamma_3$) ensured the damage-free release of wafer-scale graphene onto SiO$_2$/Si wafer (Fig. 1b, c, Supplementary Fig. 5) due to the weak adhesion bond strength. By comparison, the use of thermal release tape with larger surface energy and sticky surface as a self-supporting layer led to uncontrolled release of the wafer-scale graphene with poor macroscopic and microscopic integrity (Supplementary Fig. 6), indicating that the gradient surface energy is the key to the successful adhesion and release of wafer-scale 2D materials during transfer.

The design of gradient surface energy (GSE) allowed successful integration of the 4-inch single-crystal graphene onto the SiO$_2$/Si wafer with high intactness (99.8 ± 0.2%, Fig. 1d, e, Supplementary Figs. 7 and 8). The GSE-transferred graphene also exhibited a clean surface with significantly reduced polymer residues when compared to conventional PMMA-transferred graphene[17,22] (Fig. 1f, g, Supplementary Figs. 9–12), owing to the much lower adsorption energy of borneol on graphene than that of PMMA (Supplementary Fig. 11). In general, the flatness of graphene was influenced by the density of particles and wrinkles on the surface. In addition to the negligible surface particles, the GSE-transferred graphene maintained its flat morphology with few wrinkles, benefiting from the ultra-flat nature of graphene/Cu(111)/sapphire with significantly inhibited graphene wrinkles and Cu step bunches (Fig. 1h, Supplementary Figs. 13–14). In this way, a graphene film with an intact, clean, and ultra-flat surface was obtained on wafer-scale SiO$_2$/Si substrate.

The proposed GSE strategy can also be used for the graphene integration onto 4-inch industrial sapphire substrates (Supplementary Fig. 15). Meanwhile, wafer-scale graphene grown on Cu foil and h-BN could also be integrated onto SiO$_2$/Si using the GSE strategy (Supplementary Figs. 16 and 17). Similar results were obtained by using rosin as small molecule buffer layer[20], implying the versatility of the GSE method (Supplementary Fig. 18). To show more details of the GSE transfer method, we have included a step-by-step protocol within the Methods section, Supplementary Movie 1 and 2.

### Uniform wafer-scale graphene

The uniformity of transferred graphene is vital for advanced wafer-scale integrated devices[1–4,46]. Owing to the intact and clean surface, the GSE-transferred graphene had a very uniform sheet resistance (655 ± 39 Ω sq$^{-1}$), whose standard deviation was only ~6% over the 4-inch wafer (Fig. 2a). By contrast, the sheet resistance of PMMA-transferred looked inhomogeneous with a much higher standard deviation of ~22% (600 ± 132 Ω sq$^{-1}$), resulting from the uneven distribution of cracks and contaminations introduced during the transfer[20,21] (Fig. 2b). The uniformity of graphene at the microscopic level was further evaluated by Raman mapping. As shown in Supplementary Fig. 19, no D band peak was observed for GSE-transferred graphene. Also, the distribution of G-band position became remarkably narrower when compared to that of PMMA-transferred graphene (Fig. 2c, d). These observations further evidenced the advantages of the proposed GSE strategy for integrating wafer-scale damage-free and clean graphene with a uniform surface.

To examine the doping and strain level of GSE-transferred graphene, the peak positions of G band ($\omega_G$) and 2D band ($\omega_{2D}$) were extracted from the Raman spectra, and the correlation maps were

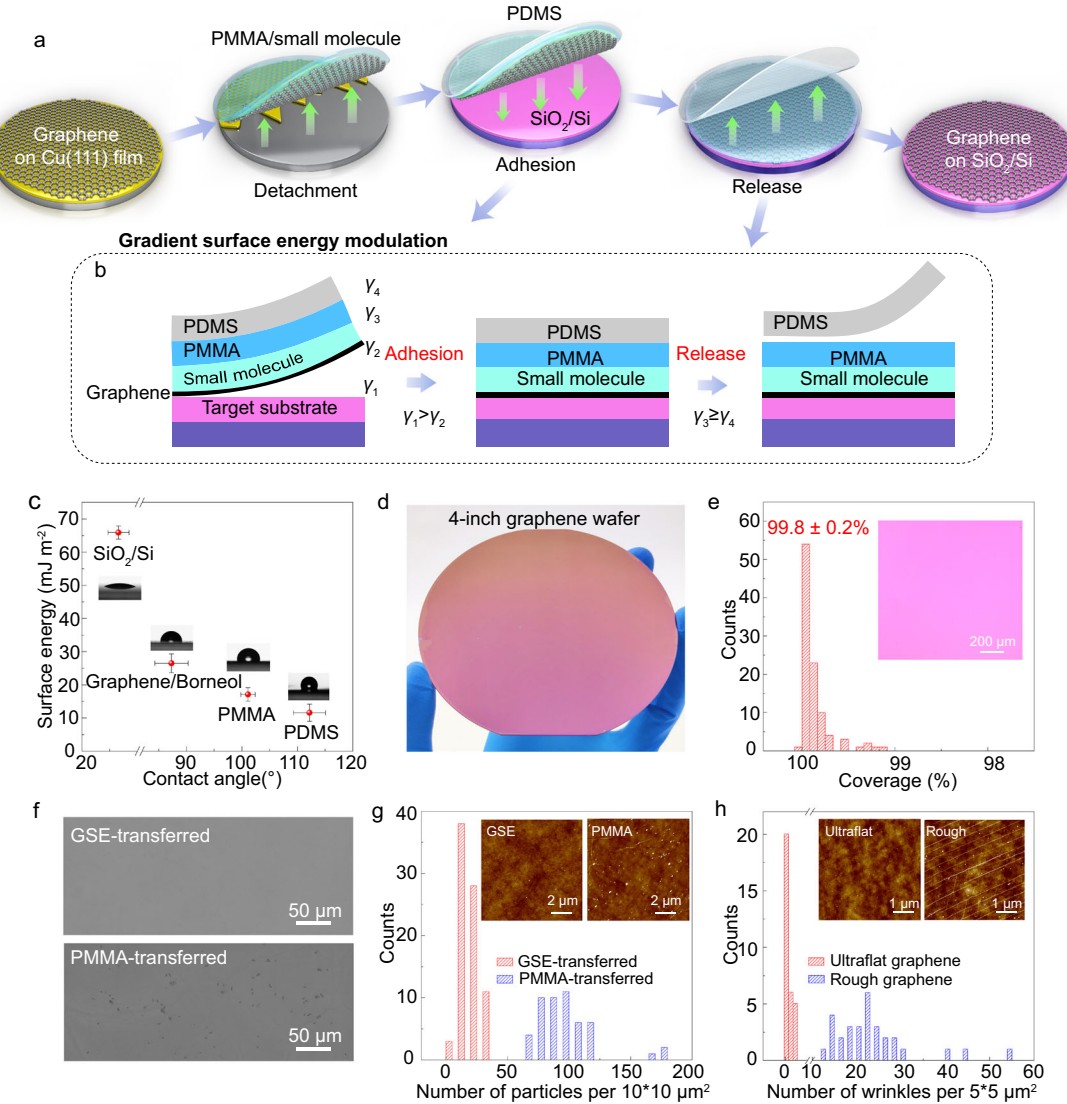

**Fig. 1 | Wafer-scale graphene integration by gradient surface energy modulation. a** Schematic illustration of wafer-scale ultra-flat graphene transfer via gradient surface energy modulation. **b** The structure of transfer medium, in which different layers with gradient surface energy are designed ($\gamma_1 > \gamma_2$, $\gamma_3 \geq \gamma_4$). Left and right figures show the adhesion and release procedures in panel (**a**). Note that the surface energy of SiO₂/Si ($\gamma_1$) is larger than that of graphene/borneol ($\gamma_2$), enabling reliable adhesion as the middle picture shows. Also, the surface energy of PDMS ($\gamma_4$) is the lowest, leading to the intact release of graphene onto the target substrate. **c** The surface energies of different surfaces calculated from measured contact angles. Insets: images showing contact angles of water on different surfaces. Error bars indicate standard deviations of surface energies and contact angles for different surface. **d** Optical image of 4-inch transferred graphene on SiO₂/Si wafer. **e** Histograms of coverage of transferred graphene. Inset: optical microscopy image of transferred graphene. **f** Scanning electron microscopy images of GSE-transferred graphene and PMMA-transferred graphene. **g** Histograms of particle number per $10 \times 10\ \mu m^2$ from 80 AFM images of GSE-transferred and 50 AFM images of PMMA-transferred graphene. Insets: Typical AFM images of GSE-transferred and PMMA-transferred graphene. **h** Histograms of wrinkle number per $5 \times 5\ \mu m^2$ issued from AFM images of transferred ultra-flat and rough graphene. Insets: AFM images of transferred ultra-flat and rough graphene.

plotted in Fig. 2e. The yellow star reveals the location of intrinsic graphene ($\omega_G = 1582\ cm^{-1}$, $\omega_{2D} = 2677\ cm^{-1}$), corresponding to neither doping nor strain[47]. The PMMA-transferred graphene films are often deeply p-doped when using wet transfer methods due to water doping at the interface[13,18,21], leading to changes in the Fermi level of graphene and declined carrier mobility. As shown in Fig. 2e, the PMMA-transferred graphene film on SiO₂/Si wafer experienced deeply p-doping. By comparison, GSE-transferred graphene on SiO₂/Si wafer nearly experienced no p-doping and strain, thereby close to the intrinsic graphene (Fig. 2e). The histograms of 2D peak's full width at half maximum ($\Gamma_{2D}$) of transferred graphene were gathered in Fig. 2f. The average $\Gamma_{2D}$ of GSE-transferred graphene (~30 cm⁻¹) was smaller than that of PMMA-transferred graphene (~40 cm⁻¹), indicating GSE-transferred graphene with little random strain fluctuation and potentially high charge carrier mobility[33,48,49].

## Electronic properties of GSE-transferred graphene

The electrical performances of devices fabricated with GSE-transferred graphene were investigated. Hall-bar devices were fabricated with standard electron beam lithography (EBL) to measure the field-effect carrier mobility of graphene on SiO₂/Si. The typical transfer characteristics of 60 Hall-bar devices fabricated with GSE- and PMMA-transferred graphene are summarized in Fig. 3a. The Dirac point of GSE-transferred graphene was near zero, and the carrier concentration was about $3 \times 10^{11}\ cm^{-2}$, revealing a very small doping level of graphene consistent with the Raman results. The extracted hole mobility reached up 10,000 cm² V⁻¹ s⁻¹ (Fig. 3a), comparable to previously reported values of state-of-the-art CVD graphene[5,16,21]. By contrast, the Dirac point of deeply-doped PMMA-transferred graphene was close to 35 V, and the carrier concentration (~$3 \times 10^{12}\ cm^{-2}$) was an order of magnitude higher than that of GSE-transferred graphene, showing a

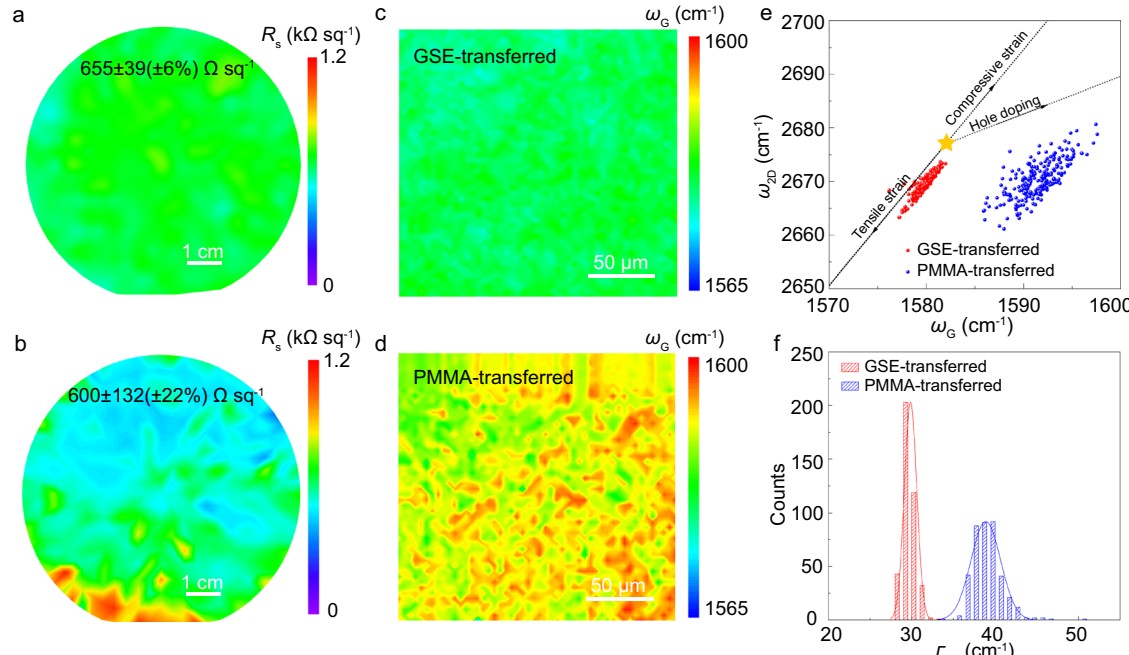

**Fig. 2 | Uniformity of transferred graphene. a, b** Spatial sheet resistance maps of GSE-transferred graphene (**a**) and PMMA-transferred graphene (**b**) on 4-inch SiO₂/Si wafer. Note that the sheet resistance deviation of GSE-transferred graphene is ~6%. **c, d** Spatial G-peak's position maps of GSE-transferred (**c**) and PMMA-transferred (**d**) graphene. The peak position of G-band of PMMA-transferred graphene is blue shifted. **e** Correlation map of the Raman G and 2D peak positions of GSE- and PMMA-transferred graphene. A total of 225 Raman spectra were taken from each type of transferred graphene. The yellow star represents the G and 2D peak positions of the pristine graphene with neither doping nor strain. **f** 2D peak's full width at half maximum ($\Gamma_{2D}$) of GSE- and PMMA-transferred graphene. The solid lines are fitting lines of distribution function for $\Gamma_{2D}$.

relatively low $\mu$ of 1930 cm² V⁻¹ s⁻¹. Accordingly, the average hole mobility of GSE-transferred graphene reached ~6000 cm² V⁻¹ s⁻¹ (Fig. 3b), a value 3-fold higher than that of PMMA-transferred graphene (~2000 cm² V⁻¹ s⁻¹). In addition, the mobility of wet-transferred graphene by only using PMMA/borneol as the transfer medium is ~3950 cm² V⁻¹ s⁻¹, much lower than that of GSE-transferred graphene, which indicate the water-adsorption-induced doping will significantly degrade the electrical properties of graphene (Supplementary Fig. 20).

As shown in Fig. 3c, the Hall mobility of the GSE-transferred graphene on SiO₂/Si extracted from the Hall effect measurement was 9500 cm² V⁻¹ s⁻¹ at room temperature. By measuring the Hall resistance ($R_{xy}$) and magnetoresistance ($R_{xx}$) at different temperatures (Fig. 3c, Supplementary Fig. 21a), we confirmed that the nonlinearity in the large magnetic field at room temperature was caused by the quantum Hall effect (QHE), further demonstrating the outstanding electrical performances and little scattering centers of GSE-transferred graphene[50] (Supplementary Fig. 22). The Hall mobilities and carrier concentrations at different temperature was shown in Supplementary Fig. 21b, and Hall mobility can reach up 19,500 cm² V⁻¹ s⁻¹ at 2 K.

To further confirm the intrinsic mobility of transferred graphene, hexagonal boron nitride (h-BN) flake was employed to pick up and encapsulate the transferred graphene from SiO₂/Si substrate (Fig. 3d) to fabricate Hall-bar devices with 1D edge contact (inset of Fig. 3e). The Hall mobility at 300 K was calculated as ~58,000 cm² V⁻¹ s⁻¹, and the carrier concentration was about $8.4 \times 10^{10}$ cm⁻², indicating excellent electrical properties of transferred graphene (Fig. 3e). According to the longitudinal magnetoresistance and Hall curve at 1.7 K (Fig. 3f), the extracted Hall mobility reached as high as 280,000 cm² V⁻¹ s⁻¹, thus rivaling mechanically exfoliated graphene[51,52]. In the Longitudinal ($R_{xx}$) and Hall ($R_{xy}$) magnetoresistance measurements at 1.7 K with a fixed magnetic field (**B** = 8.5 T), quantized Hall platform and magnetoresistance zeros were observed at all possible integer fillings of $n = 0$ and $n = 1$ Landau Levels (LLs) (Fig. 3g). Furthermore, quantization of $R_{xy}$ to

$(1/v)h/e^2$ with minimum $R_{xx}$ at fractional filling factors $v = 2/3$ and $4/3$ in $n = 0$ LL, as well as $v = 7/3$, $8/3$, $10/3$, and $11/3$ in $n = 1$ LL were observed, confirming the fractional quantum Hall effect (FQHE) feature. The back gate ($V_g$) dependent resistant measurements at the different magnetic fields on h-BN-encapsulated transferred graphene Hall-bar device were then performed to resolve the broken Landau level degeneracy and FQHE in the Landau fan diagram (Fig. 3h). The observation of FQHE further demonstrated the mobility of GSE-transferred graphene should be comparable to the high-quality exfoliated graphene[53,54] with average mobilities exceeding 100,000 cm² V⁻¹ s⁻¹. These data confirmed the ultrahigh quality of the as-transferred graphene.

**Wafer-scale integrated graphene thermal emitter devices**

Graphene-based silicon-chip blackbody emitters in the near-infrared region, including telecommunication wavelength hold promise in applications in on-Si-chip, small footprint, and high-speed emitters of highly integrated optoelectronics and silicon photonics[55]. As shown in Fig. 4a, the passage of a current $I$ through graphene thermal emitter device with narrow constrictions led to enhancements in the Joule heating, as well as localized light emission of graphene at the middle of the constriction[56]. As shown in Fig. 4b, the integrated graphene emitter device arrays with 4-inch GSE-transferred graphene wafer were successfully obtained. The representative device array was enlarged in Fig. 4c, showing an array of 8 × 8 graphene emitter devices with a graphene channel length of 120 μm and width of 10 μm at the center of graphene.

To protect the graphene channel of the thermal emitter device, a ~70-nm-thick Al₂O₃ layer was deposited on the graphene before voltage application (Supplementary Fig. 23a). Under continuous DC bias voltage, significant emission from Al₂O₃-capped graphene device between two electrodes was observed with an Infrared (IR) camera under vacuum at power density $P = 3.0$ kW cm⁻² (Fig. 4d). Note that the blue and yellow dashed lines indicated the graphene and metal

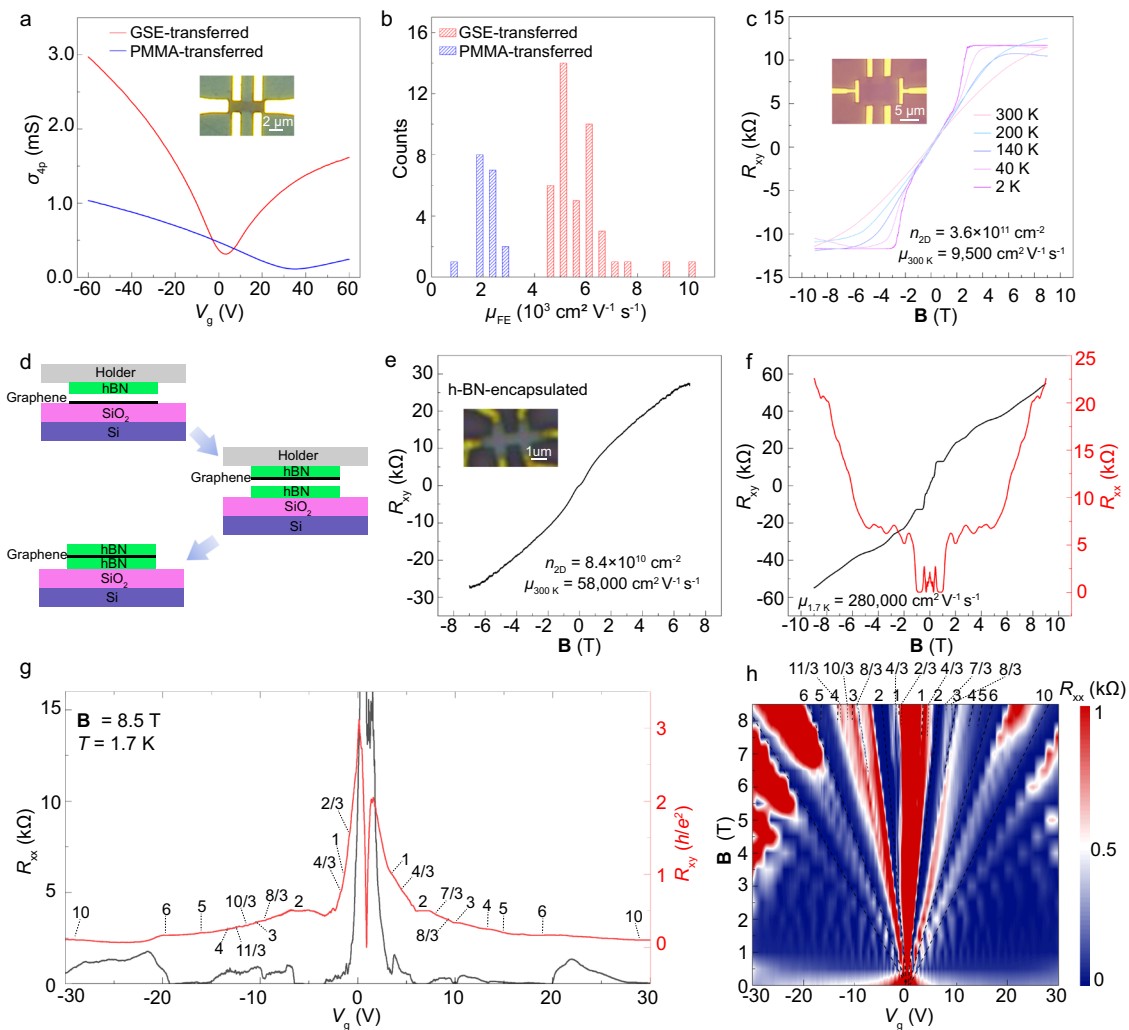

**Fig. 3 | Electrical properties of transferred graphene. a** Transfer characteristics comparison of two typical Hall-bar devices fabricated with PMMA- and GSE-transferred graphene. Inset: image of graphene Hall-bar device on SiO₂/Si. **b** Histograms of FET mobility of GSE- and PMMA-transferred graphene. The average mobility of 42 GSE-transferred and 18 PMMA-transferred graphene devices are 6000 cm² V⁻¹ s⁻¹ and 2000 cm² V⁻¹ s⁻¹, respectively. **c** The change in Hall resistance as a function of magnetic field of GSE-transferred graphene on SiO₂/Si at different temperature. Inset: image of graphene Hall-bar device on SiO₂/Si. **d** The fabrication scheme of h-BN-encapsulated transferred graphene. **e** The change in Hall resistance as a function of magnetic field (**B**) of h-BN-encapsulated transferred graphene at 300 K. Inset: image of h-BN encapsulated graphene Hall-bar device. **f** The variation in Hall resistance ($R_{xy}$) (left axis) and longitudinal resistance ($R_{xx}$) (right axis) as a function of **B** at 1.7 K. **g** $R_{xx}$ (left axis) and $R_{xy}$ (right axis) as a function of the back gate ($V_g$) at 1.7 K and 8.5 T. **h** 2D contour plot of $R_{xx}$ as a function of **B** and $V_g$. The dash black lines show LLs at filling factors $\nu = \pm2, \pm6,$ and $\pm10$, as well as some new emerging fractional filling factors $\nu = 2/3, \pm4/3, 7/3, \pm8/3...$, due to the degeneracy lifting of LLs.

electrode, respectively. The emission from these devices induced a broad spectrum in NIR region, including telecommunication wavelength. In addition, the emission intensity increased with the applied voltage (Fig. 4e). The graphene lattice temperature obtained from the peak position shift of 2D band of biased graphene[57] (Supplementary Fig. 23b) depicted a linear change with the applied power density due to the Joule heating effect[58] that reached ~750 K under vacuum at the power density of 7.7 kW cm⁻² (Fig. 4f). Thus, the graphene-based emitter is promising for high-density emitters on silicon chips, and the GSE-transferred graphene integration strategy will provide a key to the fabrication of wafer-scale graphene thermal emitter devices.

## Discussion

A general method was successfully developed for the wafer-scale graphene integration onto silicon wafers, compatible with current semiconductor technologies. The physical adhesion model and data revealed the importance of the gradient surface energy in the transfer of wafer-scale graphene, enabling reliable adhesion and release during transfer. Accordingly, 4-inch damage-free graphene with preserved intrinsic properties was obtained, contributing to a uniform sheet resistance with ~6% deviation over a 4-inch area.

The transferred graphene enhanced the electrical performance due to the negligible doping level and much fewer scattering centers when compared to conventional PMMA-transferred graphene. The Hall-bar devices fabricated with graphene on SiO₂/Si exhibited small Dirac points and high carrier mobilities (up to ~10,000 cm² V⁻¹ s⁻¹), allowing the observation of quantum Hall effect (QHE) at room temperature. Fractional quantum Hall effect (FQHE) also appeared at 1.7 K in the transferred graphene encapsulated by h-BN, with mobility reaching ~280,000 cm² V⁻¹ s⁻¹. Furthermore, integrated thermal emitter arrays with the 4-inch graphene/silicon wafer illustrated significant emission with a broad spectrum in NIR region. In sum, the proposed methodology can be used as a universal approach for the integration of other intrinsic 2D materials, such as h-BN and 2D MoS₂ on the wafer level, paving the way for the development of integrated high-performance electronics and optoelectronics.

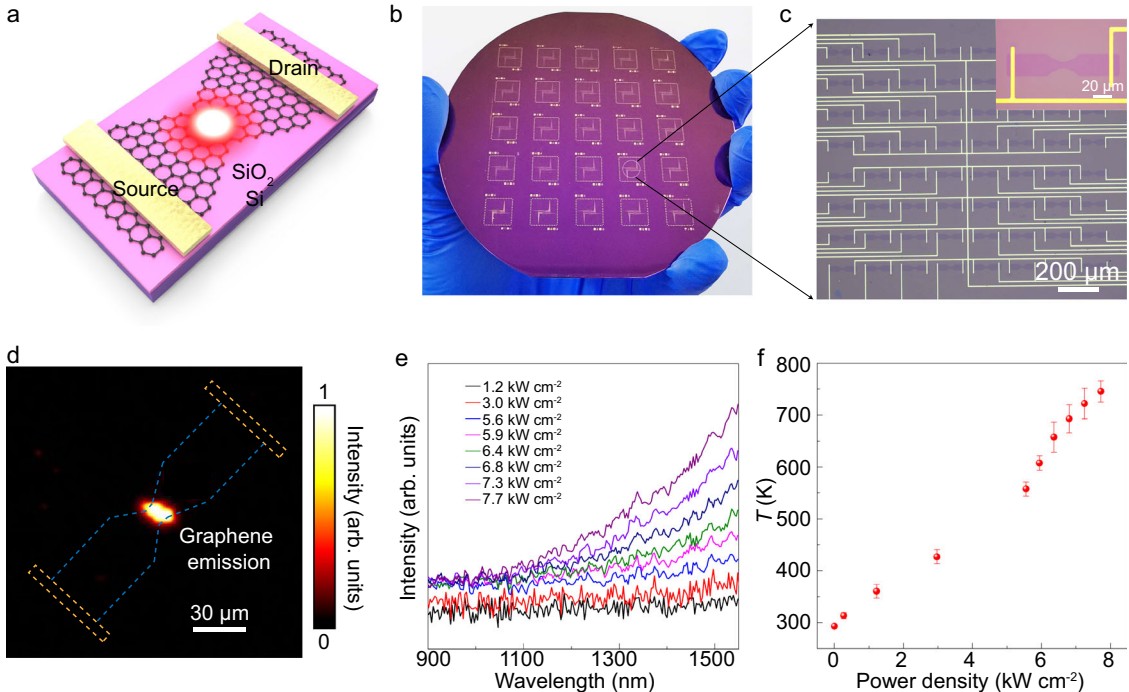

**Fig. 4 | Integration and radiation properties of wafer-scale graphene thermal emitters. a** A schematic diagram of graphene thermal emitter. **b** Wafer-scale graphene thermal emitter arrays on GSE-transferred graphene. **c** Optical microscopy image of 8 × 8 graphene thermal emitters. Inset: single graphene thermal emitter device. **d** Image of thermal emission at $P = 3.0$ kW cm$^{-2}$ captured by IR camera with false-color treatment. The bright spot in the center represents the radiation emitted by graphene. The blue dashed line and yellow dashed line indicate the graphene and metal electrodes, respectively. **e** Emission spectra issued from a graphene emitter at $P = 1.2–7.7$ kW cm$^{-2}$. **f** Graphene lattice temperatures obtained by the shift in 2D peak's position. The temperatures varied approximately linearly with power density. Error bars indicate standard deviations of temperatures at different power density.

Furthermore, our GSE transfer method was successfully reproduced by an independent research group (as shown in the Peer Review file), demonstrating the robust reproducibility of GSE method in the transfer of large-area 2D materials.

## Methods

### Preparation of single-crystal Cu(111) film on sapphire

The Cu(111) film was deposited on a single-crystal sapphire (4 inch, c plane with misorientation <0.5°, 600 μm thickness, epi-ready with $R_a < 0.2$ nm) using a sputtering system (ULVAC QAM-4W). The deposition rate was 0.3 nm s$^{-1}$ with a power of 300 W at $8 \times 10^{-4}$ Torr, and a ~500-nm-thick Cu film was obtained on sapphire after 30 min. After that, the Cu/sapphire was then annealed at 1000 °C with 2000 sccm Ar and 10 sccm H$_2$ for 2 h at atmospheric pressure in a homemade annealing furnace to obtain Cu(111) single crystal.

### Growth of ultra-flat single-crystal graphene wafers

The 4-inch Cu(111)/sapphire was heated to 1000 °C with 2000 sccm Ar at atmospheric pressure, then 40 sccm CH$_4$ (0.1% diluted in Ar) and 40 sccm H$_2$ was introduced for graphene growth. Fully covered graphene was obtained after ~2 h, and CH$_4$ gas flow was turned off while the sample cooled to room temperature.

### Preparation of single-crystal Cu(111) foil

Commercially available polycrystalline Cu foils were placed in a homemade annealing furnace equipped with a 6-inch quartz tube. Three heating zones were asynchronously heated up from room temperature to target temperatures (1040 °C, 1020 °C, 1000 °C, respectively) in 40 min and kept for 1 h, which leads to the formation of a temperature gradient through the Cu foils (about 2 °C/cm) and promotes the anomalous Cu grain growth[59]. The heating and annealing processes were carried out under 500 sccm H$_2$ and 100 sccm Ar.

### Growth of rough graphene on single-crystal Cu foil

The rough graphene was grown on Cu(111) foil using a low-pressure CVD system. Firstly, The Cu(111) foil was heated to 1000 °C with 500 sccm Ar, followed by annealing with 500 sccm H$_2$ for 30 min. Then the growth of graphene was initialed by the introduction of 1 sccm CH$_4$. Fully covered graphene was produced after 1-h growth. Finally, the system was cooled down to room temperature.

### Transfer of ultra-flat single-crystal graphene wafers with GSE strategy

A step-by-step protocol is available as a Supplementary Protocol in Supplementary Information. First, a layer of (-)-borneol (>97% purity, Alfa Aesar) dissolving in isopropyl alcohol (Crystal Clear Electronic Material Co., Ltd.) (25 wt%) and a layer of PMMA (950 K A4, Microchem Inc.) were sequentially spin-coated on graphene/Cu(111)/sapphire at 1000 rpm for 1 min, and baked at 130 °C for 3 min to form a composite support film. After that, PMMA/Borneol/graphene film was detached from growth substrate by etching the Cu film for 8-12 h in an aqueous solution of 1 mol/L (NH$_4$)$_2$S$_2$O$_8$ (Rawhn, Shanghai Yien Chemical Technology Co., Ltd.). After washing with deionized water to remove residual etchant, the PMMA/Borneol/graphene film was attached to the Si substrates, and the PDMS (WF-40×40-0060-X4, Gel-Pak) was laminated on the surface of PMMA after the graphene was dried using the commercial laminator (LM-330ID, Rayson Co., Ltd.). The composite film of PDMS/PMMA/Borneol/Graphene was detached from Si substrate in water because the water will intercalate into the interface of graphene and Si substrate due to the hydrophilic surface of SiO$_2$/Si (Supplementary Fig. 2, Supplementary Movie 1). The composite film is fully dried in atmosphere, followed by laminating onto SiO$_2$/Si (sapphire), and the PDMS was exfoliated from the PMMA at 180 °C in 5 min. To further enhance the interaction of graphene and substrate, we baked the graphene at 180 °C for 3 h

before removing PMMA and borneol with the vapor of hot acetone (UP, 99%, Crystal Clear Electronic Material Co., Ltd.), leaving the monolayer graphene on target substrate.

The detachment of graphene from Cu(111)/sapphire can also be achieved by electrochemical bubbling delamination when PDMS/PMMA/borneol was used as a composite support layer, which reduced the time of transfer of graphene and preserved the Cu(111)/sapphire wafer.

### Transfer of ultra-flat and rough single-crystal graphene wafers with PMMA

First, a layer of PMMA was spin-coated on graphene/Cu(111)/sapphire or graphene/Cu(111) foil at 1000 rpm for 1 min, and baked at 170 °C for 3 min. After that, PMMA/graphene film was detached from growth substrate by etching the Cu film in an aqueous solution of 1 mol/L $(NH)_4S_2O_8$. After washing with deionized water to remove residual etchant, the PMMA/graphene film was attached to the $SiO_2$/Si substrates at the surface of water. To further enhance the interaction of graphene and $SiO_2$/Si, the PMMA/graphene/$SiO_2$/Si was baked at 180 °C for 3 h before removing PMMA with the vapor of hot acetone, leaving the monolayer graphene on $SiO_2$/Si.

### DFT calculation of adsorption energies of borneol and PMMA on graphene surface

To evaluate the adsorption energies of borneol and PMMA on graphene surface $E_{ad}$, short chain of PMMA ($C_{22}H_{38}O_8$) terminated with $-CH_3$ without introducing additional polarity of the polymer was used in our first-principles calculations. Both borneol and PMMA short chain were placed on the surface of graphene with a ~3.3 A distance, measured from the center plane of molecule to the plane of graphene. The adsorption energy is defined as:

$$E_{ad} = E_{transfer\ medium} + E_{graphene} - E_{transfer\ medium\ on\ graphene} \quad (3)$$

All the structures were optimized using the density functional theory (DFT) implemented in VASP. To keep consistency, all the periodic cells are of the same size with $a = 24.56$ Å, $b = 24.56$ Å, $c = 21.70$ Å. The $11 \times 11$ supercell of graphene was chosen for all the calculations. The PBE with the optB86b-vdW correction functional was used to determine the total energies for each structure, taking into account the Van der Waals interaction between polymer and graphene.

### Fabrication of Hall-bar devices and electrical transport measurement

Hall-bar devices were fabricated on the graphene/$SiO_2$/Si with marks for alignment. Electron-beam lithography and plasma etching with air (Diener Pico) were employed to pattern graphene into a Hall-bar geometry. After a PMMA mask (PMMA 950 K A4 @ 4000 rpm) was patterned by EBL, Pd/Au (5/40 nm) electrodes were deposited by thermal evaporation (ZHD300, Beijing Technol Science Co., Ltd), followed by a standard lift-off technique. Limited by the stage size of electron beam lithography, we fabricated 42 graphene Hall-bar devices on 10 slices from the same 4-inch GSE-transferred graphene wafer and 18 graphene Hall-bar devices on 5 slices from another 4-inch PMMA-transferred graphene wafer. The area of each slice is $1 \times 1\ cm^2$.

Electrical transport at room temperature was determined using a vacuum-probe station (Lakeshore CRX-VF) with a semiconductor characterization system (B1500A, KeySight). Electrical transport at low temperature and magneto-transport data were acquired using a low temperature and strong magnetic electronic measurement system (AttoDry2100, Attocube). Device resistance was measured using a lock-in amplifier (Stanford Research 830) with an AC driving current of 0.1–1 μA.

### Fabrication of thermal emitters and thermal radiation measurement

The arrays of graphene thermal emitters were fabricated by UV lithography machine, and $Al_2O_3$ layer was grown by atomic layer deposition method. The Raman spectroscopy of graphene were measured by a confocal microRaman spectroscope (Renishaw inVia Qontor, UK) with a solid-state laser at 532 nm. The Raman signals were dispersed by a grating of 1800 lines/mm ensuring a high spectral resolution of ~1.0 $cm^{-1}$. The spectrometer was calibrated by a quartz tungsten lamb at temperature of 3200 K before measurement. The thermal emission spectra were recorded using a spectrometer and a liquid nitrogen cooled Si CCD with a 50× objective lens. The radiation signal was acquired using an infrared camera. All measurements were carried out at room temperature and in vacuum.

### Characterization

**Optical measurement.** Optical microscopy was conducted on a Nikon Olympus LV100ND. Raman spectra of transferred graphene were collected on a Horiba LabRAM HR Evolution Raman system using a 532 nm laser with a laser spot size 1 μm, and a 100x objective and 600 lines/mm grating were used to collect the Raman signal.

**Contact angle measurement.** The contact angle images of tested liquids on different surfaces were obtained using Biolin THETA optical tensiometer, the volume of the droplet on surface was controlled at about 4 μL.

**SEM and TEM measurement.** SEM images were obtained on an FEI Quattro S field-emission scanning electron microscope using a 5 kV acceleration voltage. The aberration-corrected STEM images of graphene were performed using a Nion U-HERMS200 microscope at 60 kV.

**AFM measurement.** The AFM morphology images were collected on a Brucker Dimension Icon using the ScanAsyst mode.

**Sheet resistance measurement.** The sheet resistances of transferred graphene on 4-inch wafers were collected by CDE ResMap 178 four-probe resistance tester.

## Data availability

The data that support the findings of this study are available within the article and its Supplementary Information files. The source data of Fig. 1c, e, g, h, 2a–f, 3a–c, e–h, 4e, f, and Supplementary Figs. 8b, 10a, 12c, 14c, 15b, d, 16b, 18c, 19a, b, 20b, c, 21a, b, and 23b are provided as "Source Data File". All raw data generated during the current study are available from the corresponding authors upon request. Source data are provided with this paper.

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

## Acknowledgements

This work was supported by the National Natural Science Foundation of China (Nos. 52021006, T2188101, 22105009, 12174444, 52072043), Beijing National Laboratory for Molecular Sciences (BNLMS-CXTD-

202001), the Tencent Foundation (XPLORER PRIZE), the National Key R&D Program of China (2018YFA0306900), the Natural Science Foundation of Hunan Province (Nos. 2020JJ3039, 2020RC3032) and the Beijing Nova Program of Science and Technology (Z191100001119067). We acknowledge Molecular Materials and Nanofabrication Laboratory (MMNL) in the College of Chemistry at Peking University for the use of instruments.

## Author contributions

H.P., X.G. and L.Z. conceived the original idea for the project. Y.C., J.T., Y.L., L.S. and Z.L. carried out the synthesis of graphene. Y.W. carried out the synthesis of h-BN. X.G., L.Z., M.Y. and G.G. carried out the transfer and characterization of graphene. L.L. carried out the DFT calculation of adsorption energy. W.W. and L.L. carried out the fabrication of h-BN-encapsulated transferred graphene. J.Q., J.W., X.G., Q.W., W.W., J.T., C.T. and J.Y. carried out the fabrication and electrical transport measurement of graphene Hall-bar devices and data analysis. F.L., M.Z. and S.Q. carried out the fabrication and measurement of graphene thermal emitters. The manuscript was written by H.P., L.Z., and X.G. with the input from other authors. The whole work was supervised by H.P. All authors contributed to the scientific planning and discussions.

## Competing interests

The authors declare no competing interests.
