## [Peer Review File · Nature Communications]

Integrated wafer-scale ultra-flat graphene by gradient surface energy modulationREVIEWER COMMENTS

Reviewer #1 (Remarks to the Author):

The noteworthy results are wafer-scale transfer of graphene with exceptional high homogeneity in electrical and Raman characteristics, as well as unusually high mobility on SiO₂ substrates. The direct performance and characteristics of the transferred graphene is impressive, as usual from this highly successful and prolific research group.

Therefore, it is tempting to assess the work as (going to have) high impact on the research field. It is also, however, possible that the consistent high quality of growth and transfer of large-area graphene, is a consequence of this particular group/collaboration to master all the steps in the complex process from growth to final electrical device better than nearly anyone else in the field.

There is nothing wrong with being excellent; the question is whether the specific results in this specific manuscript can be reproduced by other groups or industries, and if so, what is the specific difference this approach will make, in comparison with previous attempts to solve the adhesion issues. Until a scientific result is reproduced or leading to valuable learning or follow-ups, it does not really exist, in my opinion.

In short, the fact that previous literature has not demonstrated as impressive results may not be due to the superiority of the transfer method, but the superiority of the group. This is all very speculative; my point is that the high quality of the results in the manuscript does not necessarily prove that the method is significantly better than previously published methods.

The authors appear to realise this, with the cautious formulation: "However, no method has so far entirely solved these issues, and most approaches are not compatible with high-volume semiconductor 62 technology at the wafer level.", where the insertion of "entirely" reflects that this work may be less of a breakthrough and more of an incremental contribution.

The question at hand is, therefore, whether the gradient surface energy method is conceptually new, and thus could help other groups to make real progress in large-scale high-quality graphene production, as the authors correctly state has not happened yet. In other words, whether the solution presented here represents an "entire" solution, or another partial one.

The impact of this article on the field as it stands in 2022 is not clear, as there have been several papers published showing defect-free transfer of graphene
<https://www.nature.com/articles/s41467-019-08813-x>,
<https://pubs.acs.org/doi/10.1021/acsnano.0c10798> .

Specifically, there have been previous papers showing the use of small-molecule and adhesive interlayers to better control the bonding and debonding of grown 2D material to and from involved substrates <https://pubs.acs.org/doi/10.1021/acsnano.0c10798>,
<https://www.nature.com/articles/srep33096>

It is not easy for me to see the difference between the "gradient surface energy" method and inserting an intermediate layer to control the adhesion and release steps better.

This general approach is described in a review paper from 2021, which contains quite a few references, too many to cite here (<https://www.mdpi.com/2079-4991/11/11/2837/htm>). The section starts by explaining the GSE method, as far as I can see.

"4.3.3. Mechanical Methods

Mechanical methods rely on the van der Waals force between graphene and its initial

substrate being weaker than the force between graphene and either its receiving substrate or an intermediate substrate that acts as a transfer aid. These intermediate substrates have generally consisted of adhesive polymers added to the top side of the graphene, to physically pull the graphene from the substrate."

So, while the characterisation is extensive, convincing and way more comprehensive than most other publications in this genre, and the performance (mobility and uniformity) of the transferred graphene is excellent, I would like the authors to kindly explain how the GSE method is **radically** or **conceptually** different as opposed to **incrementally** different from the pre-existing literature on small-molecule and intermediate layers.

Even better it would be to show that this can be reproduced with likewise improved results at an independent lab, which, regrettably is not that common in the field. If that happens, there will only be applause from here.

Reviewer #2 (Remarks to the Author):

The authors report a new method for the production of wafer scale graphene that relies on a gradient surface energy (GSE) modulation approach to obtain a damage free, clean and ultra-flat graphene on target wafers. The transferred graphene showed very good charge carrier field effect mobility values on SiO₂/Si substrates and even the quantum Hall effect could be observed at room temperature. Finally, integrated thermal emitter arrays were fabricated on the 4-inch graphene/silicon wafers. This method could be extended to other 2D materials, transfer of h-BN is briefly demonstrated.

The ultra-flat graphene is obtained by first growing single crystal graphene on 4-inch Cu(111)/sapphire wafers and transferring it using a multifunctional trilayer (borneol/PMMA/PDMS) and multistep approach.

The manuscript is very relevant and timely for the 2D materials research community since it provides a method for the wafer scale growth and transfer of graphene with high charge carrier mobility values which is necessary to push graphene into industrial applications.

To obtain ultra-flat, damage-free and clean wafer scale graphene on target wafers is one of the greatest challenges faced by the scientific community. Specially, the intact transfer of graphene and the complete removal of the polymeric/organic residues used in the transfer process.

I would appreciate if the authors could address the following points/questions before considering it for publication:

1. There are many steps involved in the transfer process: the graphene film is deposited twice on the SiO₂/Si substrate. What happens the first-time graphene is deposited on the SiO₂/Si substrate and the PMMA/borneol is removed with acetone? What are the properties of that graphene?
2. PMMA/borneol/graphene is detached from the growth substrate by etching the Cu thin film from the sapphire, how long does this take?
3. Is the PMMA/borneol/graphene scooped onto SiO₂/Si in water the first time?
4. After laminating the PDMS onto the PMMA/borneol/graphene/SiO₂/Si stack the PDMS/PMMA/graphene is detached from the SiO₂/Si by immersing in water, can the authors expand on this?
5. For measuring the graphene/borneol contact angle (Fig. 1c), where was the stack deposited? What was the thickness of the borneol layer?
6. Fig. 1c Thickness of PMMA layer? Was it supported on a substrate?
7. Fig 1e What was the total area analysed of the 4-inch wafer? The resolution in terms

of holes is not great using the 10x magnification, for a more detailed analysis, images taken at 100x or 50x magnification (at various locations of the wafer) should be taken into consideration. What were the parameters used for the calculation of the intactness?

8. Fig 1g. How many AFM images of 10x10 μ m were taken across the 4-inch wafer?
9. What was the size of the graphene transferred used for fabricating the 60 devices?
10. Page 12 word document methods section transfer with PMMA: borneol is mentioned by mistake, right?
11. In order to confirm the cleanness of the graphene surface across the 4-inch wafer, it would be good to get more AFM statistics and use complementary techniques to determine the absence of organic residues

Reviewer #3 (Remarks to the Author):

The authors propose a transfer technique for graphene films grown on Cu(111)/sapphire, with the transferred film featured in no wrinkles, tears, negligible doping, and cleaner surface. The main idea here is to use borneol as the first contact polymer which exhibits a lower adsorption energy on pristine graphene as compared with PMMA according to their calculations in Figure S6 (supporting information). This polymer seems to play the pivotal role in their transfer technique and to achieve this technical breakthrough. If this is the case, then there will be several key questions they have to properly address before it goes for publication.

1. Why the second layer of polymer (PMMA) is needed in this transfer technique?
2. Although the PMMA is not in direct contact with graphene, it still contaminates graphene in the process of PMMA/borneol removal by rinsing in organic solvent. PMMA can be readily adsorbed on graphene and sticks tightly on the surface in the organic solvents. The authors need to provide some low-magnification TEM images to support their arguments.
3. The best way to show the cleanliness of transferred graphene is to acquire the Raman spectrum of the film in a freestanding state. Without the interference of substrate, polymer residues can be revealed near the D peak.
4. If the proposed mechanism is correct, they should be able to achieve similar results using many different kinds of polymer. Borneol is not the only choice. They should show this in the supporting information.
5. Observation of quantum Hall effect in CVD graphene is determined mainly by the crystallinity and the metal residues that are left over after wet etching of copper. It is not relevant to what they suppose to claim. On the contrary, showing QHE may mislead the readers.
6. The appearance of wrinkles after the transfer is the issue relevant to the mismatch of thermal expansion coefficient between graphene and the underlying Cu substrate. It should not disappear after graphene is lifted from the growing substrate.
7. This idea is not new. Transfer of 2D materials like TMDs which also considers the surface energy has been published in quite some papers recently.
8. Some early works for graphene transfer are not cited properly.

Response to 1st reviewer

The noteworthy results are wafer-scale transfer of graphene with exceptional high homogeneity in electrical and Raman characteristics, as well as unusually high mobility on SiO₂ substrates. The direct performance and characteristics of the transferred graphene is impressive, as usual from this highly successful and prolific research group.

Therefore, it is tempting to assess the work as (going to have) high impact on the research field. It is also, however, possible that the consistent high quality of growth and transfer of large-area graphene, is a consequence of this particular group / collaboration to master all the steps in the complex process from growth to final electrical device better than nearly anyone else in the field.

Response:

We deeply thank the reviewer's positive comments on the noteworthy results and impact of our work. The reviewer's constructive suggestions help bring significant improvements in our manuscript. We will fully address the reviewer's comments point by point in the following.

Question:

There is nothing wrong with being excellent; the question is whether the specific results in this specific manuscript can be reproduced by other groups or industries, and if so, what is the specific difference this approach will make, in comparison with previous attempts to solve the adhesion issues. Until a scientific result is reproduced or leading to valuable learning or follow-ups, it does not really exist, in my opinion.

Response:

We thank the reviewer raise this concern. In comparison with previous attempts to solve the adhesion issues, we exploited a physical adhesion model for the transfer of wafer-scale 2D materials, and revealed the gradient surface energy is a key factor for the successful adhesion and release of wafer-scale 2D materials during transfer. With this unique feature and comprehensive understanding, we developed a generic integration method to transfer wafer-scale 2D materials onto target wafers by gradient surface energy (GSE) modulation, leading to a damage-free, clean, and ultraflat graphene surface with negligible doping.

To make the transfer procedure more straightforward to the readers, we have included a step-by-step protocol within the Methods Section, Video S1, Video S2 and Figure S1 to show the details of transfer procedure for its reproducibility by other groups or industries. Actually, our GSE transfer method are recently adopted in Beijing Graphene Institute (BGI) and commonly used for the wafer-scale graphene transfer with a high yield. Furthermore, we have invited an independent lab to successfully reproduce the GSE transfer method as the reviewer suggested. A 4-inch graphene film was successfully transferred onto the SiO₂/Si wafer, exhibiting an intact, clean surface and a uniform sheet resistance over a 4-inch area (Figure R1). Beyond graphene, the wafer-scale h-BN film can be also transferred from the growth substrate onto the SiO₂/Si

wafer with the GSE method (Figure S17), demonstrating the robust reproducibility of GSE method in the transfer of large-area 2D materials.

Figure S1. The details of GSE transfer process.

[This figure has been redacted for confidentiality purposes.]

Figure R1. Wafer-scale graphene transferred with the GSE method by another independent lab. **a**, Optical images of transferred graphene. **b**, Spatial sheet resistance map of GSE-transferred graphene on 4-inch SiO₂/Si wafer. The 4-inch graphene film was transferred by the group of associate professor Qian Yang in Chongqing University of Science and Technology using GSE method.

Figure S17. Wafer-scale h-BN transferred using GSE method. a, optical image of GSE-transferred 2-inch h-BN film. **b,** Optical microscopy image of transferred h-BN with 20× objective. **c,** AFM image of transferred h-BN, yielding a small surface roughness of ~0.49 nm.

To make the transfer process more straightforward to the readers, we have revised the manuscript as following:

“...To show more details of the GSE transfer method, we included a step-by-step protocol within the Methods section, Video S1 and Video S2...” (Page 6)

“...A step-by-step protocol is available as a Supplementary Protocol in Supplementary Information...” (Page 12)

Besides, we have included the Figure S1 and Figure S17 in the Supplementary Information.

Question:

In short, the fact that previous literature has not demonstrated as impressive results may not be due to the superiority of the transfer method, but the superiority of the group. This is all very speculative; my point is that the high quality of the results in the manuscript does not necessarily prove that the method is significantly better than previously published methods.

The authors appear to realise this, with the cautious formulation: “However, no method has so far entirely solved these issues, and most approaches are not compatible with high-volume semiconductor 62 technology at the wafer level.”, where the insertion of “entirely” reflects that this work may be less of a breakthrough and more of an incremental contribution.

The question at hand is, therefore, whether the gradient surface energy method is conceptually new, and thus could help other groups to make real progress in large-scale high-quality graphene production, as the authors correctly state has not happened yet. In other words, whether the solution presented here represents an “entire” solution, or another partial one.

The impact of this article on the field as it stands in 2022 is not clear, as there have been several papers published showing defect-free transfer of graphene <https://www.nature.com/articles/s41467-019-08813-x>,

<https://pubs.acs.org/doi/10.1021/acsnano.0c10798>.

Specifically, there have been previous papers showing the use of small-molecule and adhesive interlayers to better control the bonding and debonding of grown 2D material to and from involved substrates <https://pubs.acs.org/doi/10.1021/acsnano.0c10798>, <https://www.nature.com/articles/srep33096>. It is not easy for me to see the difference between the “gradient surface energy” method and inserting an intermediate layer to control the adhesion and release steps better.

Response:

We thank the reviewer for the positive affirmation of our group. We think the impressive results in this work mainly benefit from the superiority of the GSE transfer method. After carefully reading the papers the reviewer mentioned, we believe the GSE transfer method represent a breakthrough in large-scale high-quality graphene integration. The detailed discussion will be presented as below:

(1) *Nat. Commun.* **2019**, *10*, 867. (<https://www.nature.com/articles/s41467-019-08813-x>). W. S. Leong et al. reported a paraffin-assisted transfer method to achieve the wrinkle-reduced and clean graphene (Figure R2). The paraffin-transferred graphene has smooth morphology and excellent electrical performances. This transfer method was reproduced by the group of Prof. R. S. Ruoff to transfer fold-free graphene film recently (M. Wang et al. *Nature* **2021**, *596*, 519). However, the paraffin is rather fragile to handle due to the poor mechanical strength of paraffin, which may cause severe cracks during the wafer-scale graphene transfer. Meanwhile, the small-molecule-enabled transfer is often a wet transfer method, which might lead to water doping at the interface of graphene and target substrate.

Figure R2. Paraffin-enabled graphene transfer. **a**, Schematics showing the process of paraffin-enabled graphene transfer. **b**, Schematics of the effects of paraffin’s thermal expansion on graphene wrinkle. **c-d**, Typical AFM images of graphene film transferred with PMMA (c) and paraffin (d) support layers.

(2) *ACS Nano* **2021**, *15*, 11276. (<https://pubs.acs.org/doi/10.1021/acsnano.0c10798>) Y. M. Seo et al. reported a gel-assisted direct delamination approach for the defect-free,

etchant-free, and large-area graphene transfer (Figure R3). The n-doping adhesive gel enables the mechanical exfoliation of graphene from copper foil due to the strong charge transfer interaction between graphene and polyethylenimine(PEI)-gel. The obtained graphene film on the PEI substrate is n-doping, which is suitable for the specific application such as transparent conductive film or barrier film. Unfortunately, the approach is not so compatible with semiconductor technology where electronic devices are fabricated on the industrial wafers. And the gel-induced doping might act as extra scattering centers and decrease graphene device performance.

Figure R3. Defect-free mechanical graphene transfer using n-doping adhesive gel buffer. a, Schematic illustration of a PEI-GA gel-assisted graphene transfer process. **b,** Schematic illustration of the entire transfer process. **c,** A plot of WVTR vs the optical transmittance of the transferred graphene compared with that of previously reported graphene.

(3) *Sci. Rep.* **2016**, *6*, 33096. (<https://www.nature.com/articles/srep33096>) H. V. Ngoc et al. transferred various 2D materials onto arbitrary target substrate using PMMA/PVA as transfer medium. The PVA is water-soluble, so the PMMA can be direct detached from the surface of 2D materials in the water (Figure R4). The obtained graphene exhibited a clean surface with minimum contaminants. However, this wet transfer method would lead to inevitable water-adsorption-doping and decrease graphene device performance. The average electron mobility of PMMA/PVA transferred graphene was only $\sim 3587 \text{ cm}^2 \text{ V}^{-1} \text{ s}^{-1}$. And the gate hysteresis of graphene FET device is still obvious (Figure R4e), mainly due to the water-adsorption-doping at the interface of graphene and substrate as stated in this article. Moreover, the wrinkle was not addressed and dense wrinkles were observed on the graphene surface (Figure R4d).

Figure R4. PMMA-etching-free transfer of graphene using PMMA/PVA as transfer medium. **a**, Schematic illustration of PVA assisted transfer of graphene. **b** and **d**, AFM images of transferred graphene using PMMA (**a**) and PMMA/PVA (**d**), respectively. **c** and **e**, Resistivity *versus* the back gate voltage of PMMA- (**c**) and PMMA/PVA-transferred (**e**) graphene.

In contrast, our GSE transfer strategy provides a comprehensive solution to produce the wafer-scale, damage-free, clean, and flat graphene surface with negligible doping. We found the gradient surface energy (GSE) is critical for securing the wafer-scale 2D materials integration. The conventional transfer of wafer-scale graphene with thermal release tape (TRT) often suffers from uncontrolled adhesion and release, leading to severe macroscopic and microscopic cracks in graphene (Figure R5a-b). While in our GSE method, we exploited a physical adhesion model to reveal the fracture strength of interface σ_f is proportional to the surface energy ratio of the adherend to the adhesive (γ_B/γ_A), and the design of “PDMS/PMMA/small molecules/graphene/target substrate” with gradient surface energy ($\gamma_{\text{pdms}} \leq \gamma_{\text{pmma}} < \gamma_{\text{molecules-G}} < \gamma_{\text{sub}}$) ensures reliable adhesion and release, contributing to successful 4-inch graphene transfer (Figure R5c-d). In this aspect, the GSE transfer method is conceptually different from with all the previous attempts.

Moreover, the small molecule works as a buffer layer to prevent direct contamination caused by the upper PMMA layer; and PDMS layer is robust to handle and serves as a self-supporting layer, allowing dry transfer of graphene and preventing water-adsorption-induced p-doping. Thus, the GSE transfer method can concurrently produce wafer-scale, damage-free, clean, and flat graphene surface with negligible doping (Figure R5e-h). In this aspect, we think the GSE method is also a significant development over previous attempt.

Since we have provided step-by-step protocol and supplementary videos to show the details of GSE transfer method as the reviewer suggested, we believe the new GSE transfer method will help other groups to make real progress in large-scale high-quality graphene production.

Figure R5. GSE transfer method. **a**, Optical picture of TRT/PMMA/Borneol-transferred graphene after removing TRT. **b**, Optical microscopy (OM) images of PMMA/Borneol/graphene on SiO₂/Si. **c**, Optical picture of GSE-transferred graphene after removing PDMS. **d**, OM images of PMMA/borneol/graphene on SiO₂/Si. **e**, Optical image of 4-inch transferred graphene on SiO₂/Si wafer. **f**, Histograms of coverage of transferred graphene. Inset: optical microscopy image of transferred graphene. **g**, Histograms of particle number per 10×10 μm² from 130 atomic force microscopy (AFM) images of GSE-transferred and PMMA-transferred graphene. Insets: AFM images of GSE-transferred and PMMA-transferred graphene. **h**, Histograms of wrinkle number per 5×5 μm² issued from 60 AFM images of transferred ultra-flat and rough graphene. Insets: AFM images of transferred ultra-flat and rough graphene.

Besides, we have cited the papers the reviewer mentioned as following:

27 Van Ngoc, H. et al. PMMA-etching-free transfer of wafer-scale chemical vapor deposition two-dimensional atomic crystal by a water soluble polyvinyl alcohol polymer method. *Scientific Reports* **6**, 33096 (2016).

31 Seo, Y.-M. et al. Defect-free mechanical graphene transfer using n-doping adhesive gel buffer. *ACS Nano* **15**, 11276-11284 (2021).

Question:

This general approach is described in a review paper from 2021, which contains quite a few references, too many to cite here. The section starts by explaining the GSE method (<https://www.mdpi.com/2079-4991/11/11/2837/htm>), as far as I can see.

“4.3.3. Mechanical Methods

Mechanical methods rely on the van der Waals force between graphene and its initial substrate being weaker than the force between graphene and either its receiving substrate or an intermediate substrate that acts as a transfer aid. These intermediate

substrates have generally consisted of adhesive polymers added to the top side of the graphene, to physically pull the graphene from the substrate.”

So, while the characterization is extensive, convincing and way more comprehensive than most other publications in this genre, and the performance (mobility and uniformity) of the transferred graphene is excellent, I would like the authors to kindly explain how the GSE method is **radically** or **conceptually** different as opposed to **incrementally** different from the pre-existing literature on small-molecule and intermediate layers.

Response:

We deeply appreciate the kind and insightful comments from reviewer. The “Mechanical Methods” the reviewer mentioned mainly focused on the “graphene delamination from growth substrate”. The van der Waals force between graphene and growth substrate is weaker than the force between graphene and transfer medium, so the graphene film can be direct exfoliated from the growth substrate (Figure R6a). While the GSE transfer method was designed to ensure the “reliable adhesion and release of graphene to the target substrate” (Figure R6b). The surface energy of target substrate (γ_1) is larger than that of graphene/small molecule (γ_2), enabling reliable adhesion. Besides, the surface energy of PDMS (γ_4) is lowest, leading to the intact release of wafer-scale graphene onto the target substrate. Therefore, the GSE transfer method is also conceptually different from the mechanical methods mentioned in the review (<https://www.mdpi.com/2079-4991/11/11/2837/htm>).

For the wafer-scale graphene transfer, the reliable adhesion and release of graphene film is very crucial, which determine the integrity of transferred wafer-scale graphene. Beyond graphene, the GSE methodology can be used as a universal approach for the transfer of other intrinsic 2D materials, such as wafer-scale h-BN film (Figure S17). Furthermore, we also demonstrated the integration of graphene thermal emitter arrays on the 4-inch graphene/SiO₂/Si wafer. We hope the proposed methodology here will pave the way of 2D materials transfer and integration of high-performance electronics in the field.

Figure R6. Difference between mechanical method and GSE method. a, Schematic of Mechanical exfoliation of graphene from growth substrate. **b,** Adhesion and release of graphene onto target substrate using GSE method.

To make it clear to the readers, we have revised the manuscript as following:

“...For the wafer-scale graphene transfer, both the reliable adhesion and release of graphene film are critical, which determine the integrity of transferred wafer-scale graphene...” (Page 5)

“...indicating that the gradient surface energy is the key to the successful adhesion and release of wafer-scale 2D materials during transfer...” (Page 5)

Besides, we have revised the Fig. 1b and corresponding caption as following:

“...**b**, The structure of transfer medium, in which different layers with gradient surface energy are designed. Left and right figures show the adhesion and release procedures in panel (a). Note that the surface energy of SiO₂/Si is larger than that of graphene/borneol, enabling reliable adhesion as the middle picture shows. Also, the surface energy of PDMS is the lowest, leading to the intact release of graphene onto the target substrate ...” (Page 21)

We also cited the paper the reviewer mentioned as following:

41 Langston, X. et al. Graphene transfer: A physical perspective. *Nanomaterials* **11**, 2837 (2021).

Question:

Even better it would be to show that this can be reproduced with likewise improved results at an independent lab, which, regretfully is not that common in the field. If that happens, there will only applaus from here.

Response:

We deeply appreciate the constructive suggestion from reviewer. We think the reviewer’s suggestion is important and will make a change in the field. Thus, we have invited an independent research group to reproduce our work. The GSE transfer method has been successfully reproduced by the group of associate professor Qian Yang in Chongqing University of Science and Technology. The as-obtained wafer-scale graphene exhibited an intact and clean surface, resulting in uniform sheet resistance with ~15% deviation over a 4-inch area (Figure R1). Actually, the GSE method has been well reproduced by the process engineers in Beijing Graphene Institute (BGI) for the wafer-scale graphene transfer with a high yield.

To help more researchers to produce large-area and high-quality 2D materials, a step-by-step protocol in supplementary information and Video S1 have been included to

show the details of GSE transfer method. In addition, GSE method can also be used to transfer graphene grown on Cu foil, which was also recorded in Video S2. The obtained graphene showed a uniform sheet resistance with a small deviation as shown in Figure S16.

To further demonstrate the versatility of our GSE method, wafer-scale h-BN film was also transferred from growth substrate onto SiO₂/Si via GSE. The GSE-transferred 2-inch h-BN had a clean and intact surface as shown in Figure S17.

Figure S16. Transfer of graphene grown on Cu foil onto SiO₂/Si. a, Optical image of GSE-transferred graphene on SiO₂/Si. **b,** Spatial sheet resistance map of transferred graphene with a ~7% deviation over a 4-inch area.

To make the transfer process more direct to the readers, we have revised the manuscript as following:

“...To show more details of the GSE transfer method, we included a step-by-step protocol within the Methods section, Video S1 and Video S2...” (Page 6)

“...Meanwhile, wafer-scale graphene grown on Cu foil and h-BN could also be integrated onto SiO₂/Si using the GSE strategy (Fig. S16, S17)...” (Page 6)

“...A step-by-step protocol is available as a Supplementary Protocol in Supplementary Information...” (Page 12)

Besides, we have included the Figure S16 and Figure S17 in the Supplementary Information.

Response to the 2nd Reviewer

The authors report a new method for the production of wafer scale graphene that relies on a gradient surface energy (GSE) modulation approach to obtain a damage free, clean and ultra-flat graphene on target wafers. The transferred graphene showed very good charge carrier field effect mobility values on SiO₂/Si substrates and even the quantum Hall effect could be observed at room temperature. Finally, integrated thermal emitter arrays were fabricated on the 4-inch graphene/silicon wafers. This method could be extended to other 2D materials, transfer of h-BN is briefly demonstrated.

The ultra-flat graphene is obtained by first growing single crystal graphene on 4-inch Cu(111)/sapphire wafers and transferring it using a multifunctional trilayer (borneol/PMMA/PDMS) and multistep approach.

The manuscript is very relevant and timely for the 2D materials research community since it provides a method for the wafer scale growth and transfer of graphene with high charge carrier mobility values which is necessary to push graphene into industrial applications.

To obtain ultra-flat, damage-free and clean wafer scale graphene on target wafers is one of the greatest challenges faced by the scientific community. Specially, the intact transfer of graphene and the complete removal of the polymeric/organic residues used in the transfer process.

I would appreciate if the authors could address the following points/questions before considering it for publication:

Response:

We deeply appreciate the positive and insightful comments from the reviewer on the novelty and importance of our work. The reviewer's constructive suggestions help bring significant improvements in our manuscript. We will fully address the reviewer's comments point by point in the following.

Question 1:

There are many steps involved in the transfer process: the graphene film is deposited twice on the SiO₂/Si substrate. What happens the first-time graphene is deposited on the SiO₂/Si substrate and the PMMA/borneol is removed with acetone? What are the properties of that graphene?

Response:

We appreciate the insightful and kind comments by reviewer. During the transfer process, the PMMA/borneol/graphene composite film was first scooped on Si substrate, a step made it possible to laminate the PDMS layer onto the PMMA/borneol layer.

If the PMMA/borneol was removed with acetone instead of laminating PDMS, we can also get intact and clean graphene as shown in Fig. S20a. However, the doping level of that graphene was still larger than GSE-transferred mainly due to the water-adsorption-induced doping at the interface when graphene was scooped on substrate in water (Fig.

S20b). The electrical performances of devices fabricated with such graphene films were also investigated. The typical transfer characteristics of 5 Hall-bar devices fabricated with this graphene are summarized in Fig. S20c. The Dirac point of graphene was close to 29 V, and the carrier concentration was relatively large ($\sim 2 \times 10^{12} \text{ cm}^{-2}$), showing a low carrier mobility of $\sim 3,950 \text{ cm}^2 \text{ V}^{-1} \text{ s}^{-1}$. The water at the interface deeply increased the doping level of graphene and decreased the electrical performance of graphene devices. Therefore, PDMS layer is used and serves as a self-supporting layer, allowing the dry transfer of wafer-scale graphene and preventing water-adsorption-induced doping.

Figure S20. Properties of PMMA/borneol-transferred graphene. **a**, Typical optical microscopy image of PMMA/borneol-transferred graphene on SiO₂/Si with a 20× objective. **b**, Correlation map of the Raman G and 2D peak positions of PMMA/borneol-transferred graphene comparing to GSE- and PMMA-transferred graphene. **c**, Transfer characteristics of 5 typical Hall-bar devices fabricated with PMMA/borneol-transferred graphene.

To make it clear to the readers, we have revised the manuscript as following:

“...In addition, the mobility of wet-transferred graphene by only using PMMA/borneol as the transfer medium is $\sim 3,950 \text{ cm}^2 \text{ V}^{-1} \text{ s}^{-1}$, much lower than that of GSE-transferred graphene, which indicate the water-adsorption-induced doping will significantly degrade the electrical properties of graphene (Fig. S20)...” (Page 8)

Moreover, we have included the Figure S20 and corresponding discussion in the Supplementary Information.

Question 2:

PMMA/borneol/graphene is detached from the growth substrate by etching the Cu thin film from the sapphire, how long does this take?

Response:

We are very thankful for the reviewer's comment. The chemical etching process was completed overnight (8~12 h) for 4-inch graphene/Cu(111)/sapphire wafer using 1 M (NH₄)₂S₂O₈ solution.

In addition, the detachment time can be shortened to ~ 5 min by electrochemical

bubbling method, where PDMS/PMMA/borneol was directly used as a composite support layer.

To make it more clear to the readers, we have revised the Methods section of manuscript as following:

“...PMMA/Borneol/graphene film was detached from growth substrate by etching the Cu film for 8~12 h in an aqueous solution of 1 mol/L (NH)₄S₂O₈...”

Question 3:

Is the PMMA/borneol/graphene scooped onto SiO₂/Si in water the first time?

Response:

We are very thankful for the reviewer's comment. The PMMA/borneol/graphene was scooped onto SiO₂/Si substrate in the deionized water. The details of transfer process were shown in the Video S1 and Figure S1.

Question 4:

After laminating the PDMS onto the PMMA/borneol/graphene/SiO₂/Si stack, the PDMS/PMMA/graphene is detached from the SiO₂/Si by immersing in water, can the authors expand on this?

Response:

We appreciate the insightful comment by reviewer. The water will intercalate into the interface of graphene and SiO₂/Si due to the hydrophilic surface of SiO₂/Si when PDMS/PMMA/borneol/graphene/Si immersed in water. Then the composite film can be peeled off from substrate as shown in Figure S2 and Video S1. The PDMS of obtained composite film can serve as a self-supporting layer, allowing the dry transfer of graphene to versatile wafers and preventing water doping.

Figure S2. Details of GSE transfer process. The lamination of PDMS and detachment of PDMS/PMMA/borneol/graphene composite film. The detachment was assisted by water intercalation.

To make the transfer process clear to the readers, we have revised the manuscript as following:

“...To show more details of the GSE transfer method, we included a step-by-step

protocol within the Methods section, Video S1 and Video S2...” (Page 6)

“...A step-by-step protocol is available as a Supplementary Protocol in Supplementary Information...” (Page 12)

“...detached from Si substrate in water because the water will intercalate into the interface of graphene and Si substrate due to the hydrophilic surface of SiO₂/Si (Fig. S2, Video S1). The composite film is fully dried in atmosphere...” (Page 12)

Besides, we have included the Figure S2 and corresponding discussion in the Supplementary Information.

Question 5:

For measuring the graphene/borneol contact angle (Fig. 1c), where was the stack deposited? What was the thickness of the borneol layer?

Response:

We appreciate the comment by the reviewer. As shown in Fig. R7a, the stack “graphene/borneol” was deposited on the PMMA/PDMS. And the thickness of borneol was ~1.0 μm (Fig. R7b-c).

Figure R7. The stacked structure for measuring contact angle and thickness of borneol layer. **a**, The stacked structure for measuring contact angle of graphene/borneol. **b**, Typical AFM image showing the edge of borneol layer spin-coated on the graphene/Cu(111)/sapphire. Three lines are listed to analyze the thickness of borneol layer. **c**, Height profile of three lines in (b), and the average thickness of borneol layer is ~1.0 μm.

Question 6:

Fig. 1c Thickness of PMMA layer? Was it supported on a substrate?

Response:

We appreciate the comment. The PMMA was supported on a substrate and the stack was PMMA/borneol/graphene/SiO₂/Si for contact angle measurement, as shown in Fig. R8a. And the thickness of PMMA was ~400 nm (Fig. R8b and c).

Figure R8. The stacked structure for measuring contact angle and thickness of PMMA layer. **a**, The stacked structure for measuring contact angle of PMMA. **b**, Typical AFM image showing the edge of PMMA layer. Three lines are listed to analyze the thickness of PMMA layer. **c**, Height profile of three lines in **(b)**, and the average thickness of PMMA layer is ~400 nm.

Question 7:

Fig. 1e What was the total area analysed of the 4-inch wafer? The resolution in terms of holes is not great using the 10x magnification, for a more detailed analysis, images taken at 100x or 50x magnification (at various locations of the wafer) should be taken into consideration. What were the parameters used for the calculation of the intactness?

Response:

We appreciate the comment by reviewer. We captured 100 optical microscopy images arranged in a 10×10 array over the 4-inch graphene film. The area of image captured with a 10× objective is about 0.7 cm², so the total area of Figure 1e we analyzed is about 70 cm² which is 86% of the whole 4-inch area.

We also collected 100 optical microscopy images arranged in a 10×10 array at 100× magnification as the reviewer suggested (Fig. S8a), and the average coverage of transferred graphene is 99.6 ± 0.4% (Fig. S8b). The coverage is calculated by

$$\text{Coverage} = 1 - (\text{broken area}/\text{total area})$$

In details, the pixels of whole picture, broken area and intact area are 1.78×10⁶, 1.98×10³ and 1.778×10⁶, respectively. So, the coverage equals pixels of intact area divided by pixels of total area (Fig. S8c).

Figure S8. Analysis of micro-intactness of transferred graphene. a, Optical image of a 4-inch GSE-transferred graphene film on the SiO₂/Si wafer. **b,** The histogram of graphene coverage analyzed by 100 optical microscopy images at 100× magnification from the marked area in (a). **c,** One optical microscopy image of transferred graphene. The broken area is indicated by white dashed line. **d-i,** Typical OM images of GSE-transferred graphene with 100× objective. Note that the optical image was processed with white balance to highlight microscopic damage. The scale bar is 10 μm.

To make the statistical analysis of graphene coverage more direct to the readers, we have included the **Figure S8 and corresponding discussion** in the Supplementary Information.

Question 8:

Fig 1g. How many AFM images of 10x10 μm were taken across the 4-inch wafer?

Response:

We thank the reviewer’s comment. We collected 50 10×10 μm² AFM images over the GSE-transferred 4-inch graphene and another 50 10×10 μm² from PMMA-transferred 4-inch graphene for particle density analysis.

Question 9:

What was the size of the graphene transferred used for fabricating the 60 devices?

Response:

We thank the reviewer's comment. Limited by the stage size of electron beam lithography, we fabricated 42 graphene Hall-bar devices on 10 slices from the same 4-inch GSE-transferred graphene wafer and 18 graphene Hall-bar devices on 5 slices from another 4-inch PMMA-transferred graphene wafer. The area of each slice is $1 \times 1 \text{ cm}^2$.

To show more details of device fabrication, we have revised the Methods section of manuscript as following:

“...Limited by the stage size of electron beam lithography, we fabricated 42 graphene Hall-bar devices on 10 slices from the same 4-inch GSE-transferred graphene wafer and 18 graphene Hall-bar devices on 5 slices from another 4-inch PMMA-transferred graphene wafer. The area of each slice is $1 \times 1 \text{ cm}^2$...” (Page 14)

Question 10:

Page 12 word document, methods section transfer with PMMA: borneol is mentioned by mistake, right?

Response:

We deeply thank the reviewer's kind comment. It is a mistake and we have revised the method section of manuscript as following:

“...the PMMA/graphene/SiO₂/Si was baked at 180 °C for 3 h before removing PMMA ~~and borneol~~ with the vapor of hot acetone, leaving the monolayer graphene on SiO₂/Si...”

Question 11:

In order to confirm the cleanness of the graphene surface across the 4-inch wafer, it would be good to get more AFM statistics and use complementary techniques to determine the absence of organic residues.

Response:

We thank the reviewer's constructive comment. We have collected another 30 AFM images from 4-inch GSE-graphene wafer as the reviewer suggested, so the total number for statistics has increased to 80. The surface of GSE-transferred graphene is clean, and the particle number of GSE-transferred graphene is significantly less than that of PMMA-transferred graphene (Fig. S10).

To further investigate the cleanness of graphene surface, we have transferred graphene onto the TEM grid to prepare the suspended graphene. As the reviewer #3 suggested, acquiring the Raman spectrum of graphene in a freestanding state is a reliable way to show the cleanliness of transferred graphene. Without the interference of substrate,

polymer residues can be revealed near the D peak. As shown in Figure S12, two obvious peaks at 1330 cm^{-1} and 1430 cm^{-1} appeared in the PMMA-transferred suspended graphene (Fig. S12c), corresponding with the previous reported results (*ACS Nano* **2011**, 5, 2362). In contrast, no peaks were observed near the D band of GSE-transferred suspended graphene. Besides, the intensity ratio of 2D peak to G peak (I_{2D}/I_G) is informative of impurities on the graphene surface (*Adv. Mater.* **2017**, 29, 1700639). The I_{2D}/I_G of GSE-transferred graphene is ~ 3.5 , much higher than that of PMMA-transferred graphene (~ 1.8), indicating the clean graphene surface with few impurities.

Figure S10. Histograms of particle number and typical AFM images of GSE-transferred graphene. a, Histograms of particle number per $10 \times 10 \mu\text{m}^2$ from 80 AFM images of GSE-transferred graphene and 50 AFM images of PMMA-transferred graphene. **b-i,** Typical AFM images of GSE-transferred ultra-flat graphene. The scale bar is $2 \mu\text{m}$.

Figure S12. Raman spectra of suspended GSE- and PMMA-transferred graphene. **a**, SEM image of GSE-transferred graphene on Au TEM grid. **b**, Typical SEM image of suspended graphene membranes. **c**, The Raman spectra of GSE- and PMMA-transferred suspended graphene. Inset: fine spectra of PMMA-transferred graphene from 1100 to 1500 cm^{-1} .

To confirm the cleanness of the graphene surface across the 4-inch wafer, we have revised the Fig. 1g and corresponding caption as following:

“...**g**, Histograms of particle number per $10 \times 10 \mu\text{m}^2$ from 80 AFM images of GSE-transferred and 50 AFM images of PMMA-transferred graphene. Insets: Typical AFM images of GSE-transferred and PMMA-transferred graphene...”

Moreover, we have included the Figure S10, Figure S12 and corresponding discussions in the Supplementary Information.

Response to the 3rd Reviewer

The authors propose a transfer technique for graphene films grown on Cu(111)/sapphire, with the transferred film featured in no wrinkles, tears, negligible doping, and cleaner surface. The main idea here is to use borneol as the first contact polymer which exhibits a lower adsorption energy on pristine graphene as compared with PMMA according to their calculations in Figure S6 (supporting information). This polymer seems to play the pivotal role in their transfer technique and to achieve this technical breakthrough. If this is the case, then there will be several key questions they have to properly address before it goes for publication.

Response:

We deeply appreciate the positive and insightful comments from the reviewer on the results of our work. The reviewer's constructive suggestions help bring significant improvements in our manuscript. We will fully address the reviewer's comments point by point in the following.

Question 1:

Why the second layer of polymer (PMMA) is needed in this transfer technique?

We deeply appreciate the insightful comments from the reviewer. The borneol layer is rather fragile, so the PMMA layer is designed to support the borneol/graphene and ensures the integrity of wafer-scale graphene during transfer. If the PMMA layer is not used in the transfer medium, the obtained graphene film will suffer from dense cracks (Fig. S3).

To make the role of PMMA layer more clear to the readers, we have revised the manuscript as following:

“...The PMMA layer ensured the integrity of graphene during transfer (Fig. S3)...”
(Page 4)

Moreover, we have included the **Figure S3 and corresponding discussions** in the Supplementary Information.

Figure S3. Optical microscopy images of transferred graphene without PMMA support layer. a and b, Optical microscopy images of PDMS-transferred graphene with 10× and 100× objective, respectively.

Question 2:

Although the PMMA is not in direct contact with graphene, it still contaminates graphene in the process of PMMA/borneol removal by rinsing in organic solvent. PMMA can be readily adsorbed on graphene and sticks tightly on the surface in the organic solvents. The authors need to provide some low-magnification TEM images to support their arguments.

Response:

We deeply appreciate the insightful and constructive comments from the reviewer. We agree with the reviewer that some PMMA residues will adsorb onto the graphene surface when PMMA/borneol is removed by rinsing in the organic solvent. To avoid this, we used the hot vapor of acetone to remove PMMA/borneol by heating liquid acetone to the boiling temperature. The vapor of acetone can remove most PMMA and drop into the liquid acetone. Then, fresh acetone vapor will rise and further remove the PMMA and borneol residues on the graphene surface.

As the reviewer suggested, low-magnification TEM images of GSE- and PMMA-transferred graphene are shown in Figure R9. Many polymer residues were adsorbed on the surface of PMMA-transferred graphene (Fig. R9a), and almost no clean area was observed on the graphene surface (Fig. R9b-c). In contrast, no obvious polymer residue was observed on the GSE-transferred graphene (Fig. R9d). The cleanness of GSE-transferred graphene (Fig. R9e-f) was comparable to the mechanically exfoliated graphene (Fig. R9g-h, Ref: *Nature* **2008**, 454, 319), and the graphene lattice can be well characterized (Fig. R9f).

Figure R9. TEM images of PMMA-transferred, GSE-transferred and mechanical exfoliated graphene. a-c, PMMA-transferred graphene. d-f, GSE-transferred graphene. g-h, TEM images of exfoliated graphene from graphite (Data from *Nature* **2008, 454,**

319).

To make the transfer process more direct to the readers, we have revised manuscript as following:

“...removing PMMA and borneol with the vapor of hot acetone...” (Page 12)

We have added a step-by-step protocol of GSE transfer method in Supplementary Information as following:

“...The hot vapor of acetone was used to remove PMMA/borneol by heating liquid acetone to the boiling temperature. The vapor of acetone can remove most PMMA and drop into the liquid acetone. Then, fresh acetone vapor will rise and further remove the PMMA and borneol residues on the graphene surface...”

Besides, we have included the low-magnification TEM images and corresponding discussions in the **Figure S11** in the supplementary information:

Figure S11. Comparison of adsorption energies of borneol and PMMA and typical transmission electron microscopy (TEM) images of GSE- and PMMA-transferred graphene. a, The adsorption energies of borneol and PMMA on graphene. **b-c,** Typical low magnification TEM images of GSE- and PMMA-transferred graphene. The adsorption energy of borneol on graphene was approximately one half of that of

PMMA, leading to a cleaner surface with a clearly atomic image. **d-e**, Typical HRTEM images of GSE-transferred (**d**) and PMMA-transferred graphene (**e**).

Question 3:

The best way to show the cleanliness of transferred graphene is to acquire the Raman spectrum of the film in a freestanding state. Without the interference of substrate, polymer residues can be revealed near the D peak.

Response:

We deeply thank the reviewer's constructive comment. To confirm the absence of organic residues, we transfer graphene onto TEM grid to prepare suspended graphene membranes for Raman measurement as the reviewer suggested (Fig. S12a-b).

Two obvious peaks at 1330 cm^{-1} and 1430 cm^{-1} appeared in the PMMA-transferred suspended graphene (Fig. S12c), corresponding with the previous reported results (*ACS Nano* **2011**, 5, 2362). In contrast, no peaks were observed near the D band of GSE-transferred suspended graphene. Besides, the intensity ratio of 2D peak to G peak (I_{2D}/I_G) is informative of impurities on the graphene surface (*Adv. Mater.* **2017**, 29, 1700639). The I_{2D}/I_G of GSE-transferred graphene is ~ 3.5 , much higher than that of PMMA-transferred graphene (~ 1.8), indicating the clean graphene surface with few impurities.

To further confirm the cleanness of transferred graphene, we have included the **Figure S12 and corresponding discussions** in the Supplementary Information.

Figure S12. Raman spectra of GSE- and PMMA-transferred suspended graphene. **a**, SEM image of GSE-transferred graphene on Au TEM grid. **b**, Typical SEM image of suspended graphene membranes. **c**, The Raman spectra of GSE- and PMMA-transferred suspended graphene. Inset: fine spectra of PMMA-transferred graphene from 1100 to 1500 cm^{-1} .

Question 4:

If the proposed mechanism is correct, they should be able to achieve similar results using many different kinds of polymer. Borneol is not the only choice. They should show this in the supporting information.

Response:

We deeply thank the reviewer's constructive and insightful comment. We achieved similar results using the same GSE method by replacing borneol with rosin, a small natural organic molecule that has a weak interaction with graphene (*Nat. Commun.* **2017**, 8, 14560). As shown in Figure S18, the wafer-scale GSE-transferred graphene is intact and clean. No D band can be seen in Raman spectra (Fig. S18c), and the FWHM of 2D peak is $\sim 28 \text{ cm}^{-1}$, indicating the GSE-transferred graphene has little random strain fluctuation and potentially high charge carrier mobility.

Figure S18. GSE transfer results using rosin as a small molecule buffer layer. a, Optical image of 4-inch GSE-transferred graphene. **b,** Typical optical microscopy image of graphene at 20 \times magnification. **c,** Typical Raman spectra of GSE-transferred graphene on the SiO₂/Si wafer.

To show the versatility of GSE method, we have revised the manuscript as following:

“...Similar results were obtained by using rosin as small molecule buffer layer²⁰, implying the versatility of the GSE method (Fig. S18)...” (Page 6)

Besides, we have included the **Figure S18 and corresponding discussions** in the Supplementary Information.

Question 5:

Observation of quantum Hall effect in CVD graphene is determined mainly by the crystallinity and the metal residues that are left over after wet etching of copper. It is not relevant to what they suppose to claim. On the contrary, showing QHE may mislead the readers.

Response:

We deeply thank the reviewer's kind comment. The QHE can be observed at room temperature due to the highly unusual nature of charge carriers in graphene, which behave as massless Dirac fermions and move with little scattering under ambient conditions. There are several factors that help the QHE in graphene to survive to room temperature. One of the critical factors is ultra-high mobility, a mobility of $\sim 10,000 \text{ cm}^2 \text{ V}^{-1} \text{ s}^{-1}$ yields a scattering time of $\tau \sim 10^{-13} \text{ s}$, so that the high field limit $\omega_c \tau = \mu \cdot B \gg 1$

is reached in fields of several T (*Science*, **2007**, 315, 1379).

The Hall mobility of GSE-transferred graphene on the SiO₂/Si was very high (~9500 cm² V⁻¹ s⁻¹), because the transferred graphene film was clean, ultra-flat and with negligible doping, indicating little scattering centers. Thus, we can observe the QHE at room temperature. In comparison, Y. P. Chen et al. can only observe the QHE in PMMA-transferred CVD graphene at a very low temperature (0.6 K), which can be attributed on the lower mobility (~3000 cm² V⁻¹ s⁻¹) caused by scattering centers, such as water doping, polymer contamination and wrinkles introduced by PMMA wet transfer process (Fig. S22, *Appl. Phys. Lett.* **2010**, 96, 122106).

Beyond QHE, we also observed FQHE (Fig. 3g-h). To observe FQHE, the mobility of transferred graphene should be comparable to the high-quality exfoliated graphene with average mobilities exceeding 100,000 cm² V⁻¹ s⁻¹ (*2D Mater.* **2020**, 7, 041007). GSE-transferred graphene had a clean, ultra-flat and negligible doping surface, which had an ultra-high mobility (~280,000 cm² V⁻¹ s⁻¹) after encapsulation by h-BN, rivaling mechanical exfoliated graphene. Thus, the observation of FQHE at 8.5 T, 1.7 K indicated an outstanding quality of our GSE-transferred graphene (Fig. 3g-h).

Figure S22. Scattering factors affecting mobility of graphene. Scattering factors affecting mobility of graphene on SiO₂/Si.

To make the QHE more clear to the readers, we have revised the manuscript as following:

“...we confirmed that the nonlinearity in the large magnetic field at room temperature was caused by the quantum Hall effect (QHE), further demonstrating the outstanding electrical performances and little scattering centers of GSE-transferred graphene⁵⁰ (Fig. S22)...” (Page 8)

“...The observation of FQHE indicated the mobility of GSE-transferred graphene should be comparable to the high-quality exfoliated graphene^{53,54} with average mobilities exceeding 100,000 cm² V⁻¹ s⁻¹...” (Page 9)

Besides, we have included the Figure S22 and corresponding discussions in the Supplementary Information.

Question 6:

The appearance of wrinkles after the transfer is the issue relevant to the mismatch of thermal expansion coefficient between graphene and the underlying Cu substrate. It should not disappear after graphene is lifted from the growing substrate.

Response:

We thank the reviewer's kind and insightful comment. Graphene wrinkles and Cu step bunches are usually formed during the growth of graphene, due to the mismatch of thermal expansion coefficient between graphene and the underlying Cu substrate.

To minimize the adverse effects of wrinkles on the charge carrier mobility, ultra-flat graphene films were grown on the 4-inch Cu(111)/sapphire wafers for graphene transfer. There are few wrinkles in the graphene grown on the Cu(111)/sapphire wafer (Figure S14a-b), owing to the small thermal expansion of the Cu(111) thin film (~500 nm in thickness) on sapphire and the relatively strong interfacial coupling between Cu(111) and graphene (*ACS Nano* **2017**, *11*, 12337).

Moreover, the height of Cu step of ultraflat graphene/Cu(111)/sapphire is only ~2 nm, which is significantly smaller than that of copper foil (~20 nm) with dense Cu step bunches (Fig. S14c). We found that the step bunches of copper foil will cause the formation of new wrinkles after the transfer, revealed by our *in-situ* transfer process. As shown in Figure S14f, three folds and dense step bunches were observed on the graphene grown on the copper foil. After the transfer, the folds did not disappear, and new wrinkles appeared along the direction of step bunches (Fig. S14g). Since the step bunches were largely inhibited on the ultraflat graphene grown on the Cu(111)/sapphire, the transfer-induced wrinkles can be significantly reduced (Fig. S14d).

In summary, the GSE-transferred graphene with few wrinkle mainly benefits from the ultra-flat nature of graphene single crystal grown on Cu(111)/sapphire. And we have presented the conclusions in the section of "Design of wafer-scale graphene integration" as following:

"...In addition to the negligible surface particles, the GSE-transferred graphene maintained its flat morphology with few wrinkles, benefiting from the ultra-flat nature of graphene/Cu(111)/sapphire **with significantly inhibited graphene wrinkles and Cu step bunches** (Fig. 1h, S13-S14)..." (Page 6)

To make the reduction of wrinkles clearer to the readers, we have revised the **Figure S14** and **included the discussion** in the Supporting Information.

Figure S14. The relationship between the roughness and wrinkle density of transferred graphene. a and b, OM and AFM images of ultra-flat graphene grown on

Cu(111) film. **c**, Height profile of ultra-flat graphene on Cu film (**b**) and rough graphene on Cu foil (**f**). **d**, The AFM image of *in situ* transferred graphene in (**b**). There were few folds and step bunches in ultra-flat graphene, and the surface roughness (R_a) of graphene in (**b**) was 0.30 nm, leading to a smooth graphene surface after transfer (**d**). **e** and **f**, OM and AFM images of rough graphene grown on Cu(111) foil. **g**, The AFM image of *in situ* transferred graphene in (**f**). The surface roughness of rough graphene (**f**) was 12.1 nm with several folds (white arrows) and many Cu step bunches. Step bunches will turn into new wrinkles (red arrows) after transfer (**g**).

Question 7:

This idea is not new. Transfer of 2D materials like TMDs which also considers the surface energy has been published in quite some papers recently.

Response:

We thank the reviewer's comment. We have carefully read the papers about surface-energy-assisted transfer of 2D materials like TMDs. The surface-energy-assisted transfer methods mainly focused on the detachment of 2D TMDs from its growth substrate. One key strategy is exploiting the penetration of water molecules between the TMD films and the growth substrate to lift off the film at room temperature (Fig. R10 a-b; *ACS Nano* **2014**, 8, 11522; *Scientific Reports* **2019**, 9, 1641; *2D Mater.* **2021**, 8, 032001). Typically, the TMD films are hydrophobic, while the growth substrates are hydrophilic (Fig. R10c). The different surface energies can drive water molecules to penetrate underneath the film, and therefore the process is termed surface-energy assisted transfer.

However, to the best of our knowledge, no transfer medium with a gradient surface energy (GSE) distribution has been reported so far. And our GSE transfer method focuses on the adhesion and release of wafer-scale 2D materials onto target substrate (Fig. R10d). The design of the transfer medium can concurrently ensure both the reliable adhesion and release of 2D materials, leading to an intact, clean, transferred graphene with a low doping level. In details, the larger surface energy of target substrate ensures the complete wetting with less voids and reliable adhesion of graphene. Meanwhile, the low surface energy of PDMS ensure the intact release due to the weak adhesion force. All the factors drive the integration of wafer-scale ultra-flat graphene onto versatile substrate with intact, clean, and less-doping surface. More importantly, we exploited an adhesion model for the transfer of wafer-scale 2D materials, and revealed the difference of surface energy is a key to reliable adhesion and release of wafer-scale 2D materials during transfer.

Figure R10. Differences between TMDs transfer with surface energy modulation and GSE method. **a** and **b**, Illustrations of the water-assisted and surface-energy-assisted transfer process (Data from *ACS Nano* **2014**, *8*, 11522; *Scientific Reports* **2019**, *9*, 1641). **c**, Illustration of the water penetration process at the 2D MoS₂/SiO₂ interface, which assisted TMDs detachment from growth substrate (Data from *Scientific Reports* **2019**, *9*, 1641). **d**, Adhesion and release of graphene onto target substrate using GSE method.

To make the novelty of GSE transfer methods more direct to the readers, we have revised the manuscript as following:

“...For the wafer-scale graphene transfer, both the reliable adhesion and release of graphene film are critical, which determine the integrity of transferred wafer-scale graphene...” (Page 5)

“...indicating that the gradient surface energy is a key to the successful adhesion and release of wafer-scale 2D materials during transfer...” (Page 5)

Besides, we have revised the Fig. 1b and corresponding caption as following:

“...**b**, The structure of transfer medium, in which different layers with gradient surface energy are designed. Left and right figures show the adhesion and release procedures in panel (a). Note that the surface energy of SiO₂/Si is larger than that of graphene/borneol, enabling reliable adhesion as the middle picture shows. Also, the surface energy of PDMS is lowest, leading to the intact release of graphene onto the target substrate ...” (Page 21)

8. Some early works for graphene transfer are not cited properly.

We deeply thank the reviewer’s comment. We have cited additional important works and reviews for graphene transfer in the manuscript as the reviewer suggested.

We have accordingly revised the manuscript as following:

“...shown that the **optimization of PMMA** and replacement of PMMA with small molecules or other polymers would facilitate clean graphene transfer¹⁹⁻²⁷ conformal contact with the target substrate may reduce the formation of cracks and wrinkles²⁸⁻³¹, and the development of dry transfer methods may diminish water doping by preventing the submersion of target substrate in liquids³²⁻³⁹. However, no method has so far entirely solved these issues, and most approaches are not compatible with high-volume semiconductor technology at the wafer level^{40,41} ...”

“... ”

25 Li, X. *et al.* Transfer of large-area graphene films for high-performance transparent conductive electrodes. *Nano Lett.* **9**, 4359-4363 (2009).

26 Liang, X. *et al.* Toward clean and crackless transfer of graphene. *ACS Nano* **5**, 9144-9153 (2011).

... ”

36 Kim, K. S. *et al.* Large-scale pattern growth of graphene films for stretchable transparent electrodes. *Nature* **457**, 706-710 (2009).

37 Banszerus, L. *et al.* Ultrahigh-mobility graphene devices from chemical vapor deposition on reusable copper. *Sci. Adv.* **1**, e1500222 (2015).

... ”

40 Kang, J. *et al.* Graphene transfer: key for applications. *Nanoscale* **4**, 5527-5537 (2012).

41 Langston, X. *et al.* Graphene transfer: A physical perspective. *Nanomaterials* **11**, 2837 (2021).

... ”

REVIEWERS' COMMENTS

Reviewer #1 (Remarks to the Author):

The authors have done an excellent job of carefully replying to all comments and concerns of mine; the manuscript can be published in its present form.

Reviewer #2 (Remarks to the Author):

The authors have satisfactorily addressed all my questions/concerns. I believe that the manuscript is ready for publication in Nature Communications.

Reviewer #3 (Remarks to the Author):

As mentioned by several reviewers that using small polymer molecules as the transfer medium has been proposed by previously reported papers. I am not entirely convinced by their response to the concern of novelty. Nevertheless, the authors has indeed demonstrated an impressive transfer results based on the recipe they provide. It might be due to the fact that they have engineered for long time and thus mastered the technique. Before publication, they really have to honor the reported papers that critically show the success of current transfer approach. Some missing citations:

1. Y. C. Lin, C. C. Lu, C. H. Yeh, C. Jin, K. Suenaga, P. W. Chiu

"Graphene annealing: how clean can it be?", *Nano Letters*, 2012, 12, 414.

2. Y. C. Lin, C. Jin, J. C. Lee, S. F. Jen, K. Suenaga, and P. W. Chiu

"Clean transfer of graphene for isolation and suspension", *ACS Nano*, 2011, 5, 2362.

Response to the 3rd reviewer

As mentioned by several reviewers that using small polymer molecules as the transfer medium has been proposed by previously reported papers. I am not entirely convinced by their response to the concern of novelty. Nevertheless, the authors has indeed demonstrated an impressive transfer results based on the recipe they provide. It might be due to the fact that they have engineered for long time and thus mastered the technique.

Response:

We deeply thank the reviewer's comments. We'd like to point out that our GSE transfer method is designed to ensure the both reliable adhesion and release of wafer-scale 2D materials to the target substrate by gradient surface energy modulation, which is different with the previously reported work using small molecule as transfer medium.

Before publication, they really have to honor the reported papers that critically show the success of current transfer approach. Some missing citations:

- 1. Y. C. Lin, C. C. Lu, C. H. Yeh, C. Jin, K. Suenaga, P. W. Chiu "Graphene annealing: how clean can it be?", Nano Letters, 2012, 12, 414.*
- 2. Y. C. Lin, C. Jin, J. C. Lee, S. F. Jen, K. Suenaga, and P. W. Chiu "Clean transfer of graphene for isolation and suspension", ACS Nano, 2011, 5, 2362.*

Response:

We thank the reviewer's positive comments of our work and recommend publication. We have included the reported papers mentioned in Reference of latest manuscript.

- 17 Lin, Y. C. *et al.* Graphene annealing: how clean can it be? *Nano Lett.* **12**, 414-419 (2012).
- 22 Lin, Y. C. *et al.* Clean transfer of graphene for isolation and suspension. *ACS Nano* **5**, 2362-2368 (2011).